# Impact of *Ectropis grisescens* Warren (Lepidoptera: Geometridae) Infestation on the Tea Plant Rhizosphere Microbiome and Its Potential for Enhanced Biocontrol and Plant Health Management

**DOI:** 10.3390/insects16040412

**Published:** 2025-04-14

**Authors:** He Liu, Wei Chen, Xiaohong Fang, Dongliang Li, Yulin Xiong, Wei Xie, Qiulian Chen, Yingying You, Chenchen Lin, Zhong Wang, Jizhou Wang, Danni Chen, Yanyan Li, Pumo Cai, Chuanpeng Nie, Yongcong Hong

**Affiliations:** 1College of Horticulture, Fujian Agriculture and Forestry University, Fuzhou 350007, China; liuhe687@163.com (H.L.);; 2College of Tea and Food Science, Wuyi University, Wuyishan 354300, China; 3College of Resources and Environment, Fujian Agriculture and Forestry University, Fuzhou 350007, China

**Keywords:** *Camellia sinensis*, *Ectropis grisescens* Warren, rhizosphere microbiome, *Burkholderia*, transcriptomic analysis, metagenomic analysis

## Abstract

Rhizosphere bacteria play a key role in helping plants resist stress, but it has not been explored whether tea trees can also mobilize their rhizosphere bacteria to help them resist the stress of the *Ectropis grisescens* Warren (Lepidoptera: Geometridae) after being invaded by it. The study took samples of leaves, roots, and rhizosphere soil at different times after the plants were attacked by *E. grisescens*. Using 16S rRNA sequencing, we observed significant shifts in bacterial communities by the seventh day. Transcriptomic analysis showed that the *E. grisescens* attack triggered reprogramming of the tea root transcriptome, leading to changes in the rhizosphere bacterial community. Analysis of the rhizosphere soil on the seventh day revealed alterations in the microbial network structure and the core microorganisms within the network following the attack; microorganisms related to nitrogen (N) metabolism were mobilized in the rhizosphere. Notably, nitrogen-fixing bacteria from the genus *Burkholderia* were actively recruited and became important contributors in the rhizosphere. When these separated *Burkholderia* strains were reintroduced into tea trees, the level of insect-resistant substances in tea leaves was increased, thereby improving the insect resistance of tea trees.

## 1. Introduction

As sessile organisms constantly vulnerable to herbivore attacks, plants must adapt to their environment and respond to various stimuli throughout their development, shaping their interactions with other organisms [1]. When subjected to pest-induced stress, a plant’s defense system can be activated systemically. Not only do plants synthesize defensive compounds to counteract the stress, but they also release “cry for help” signals into the environment, attracting beneficial organisms such as predatory insects and microbes to combat the stress. For example, plants can attract predatory or parasitic wasps by changing the composition and content of specific volatile compounds in their aboveground parts [2,3,4,5,6]. Concurrently, they transmit signals to neighboring plants, preparing them for defense [2,3,4]. In addition, the underground roots of plants have evolved a similar “cry for help” strategy. By synthesizing and secreting specific root metabolites, they recruit beneficial soil microorganisms to minimize damage caused by stressors [6,7].

Certain underground microorganisms possess the ability to induce a series of physiological alterations in plants, thereby influencing the behavior and performance of herbivorous insects [8]. Among these microorganisms, rhizosphere growth-promoting bacteria (PGPR) represent a significant category that contributes to plant growth and development. Numerous strains of PGPR have been reported to enhance plant resistance to abiotic stresses, such as drought and salt stress, as well as biotic stresses, such as pathogenic microorganisms and insect pests [9,10,11]. This intricate interplay between plants and their associated microbiota forms a holobiont, which significantly affects plant fitness [12,13] and plays a crucial role in enhancing plant stress resistance.

Research has demonstrated that herbivorous insects, which feed on the aboveground parts of plants, can significantly affect the community structure [14] and colonization [15] of root rhizosphere microorganisms. In some cases, these alterations have been found to feedback and influence plant–insect interactions [16]. For example, damage caused by *Holotrichia parallela* Motschulsky (Coleoptera: Melolonthidae) larvae to peanut plants has been observed to alter the rhizosphere bacterial community structure of peanut plants [17]. Similarly, aphid infestation [*Myzus persicae* (Sulzer) (Hemiptera: Aphididae)] on pepper leaves has been shown to trigger the recruitment of root exudates and root-associated *Bacillus*, thereby increasing plant resistance to insects [18]. Aboveground leafminers [*Liriomyza trifolii* (Burgess) (Diptera: Agromyzidae)] feeding has been found to significantly reshape the cowpea rhizosphere microbiome, which responds by recruiting nitrogen metabolism-related microbes to promote plant growth and bolster plant defense against this pest [8]. Whitefly infestation [*Bemisia tabaci* (Gennadius) (Hemiptera: Aleyrodidae)] has been shown to reshape the rhizosphere microbiota structure of pepper, leading to the recruitment of fluorescent *Pseudomonas*, with proven insecticidal ability [19]. Likewise, aphid herbivory by *Macrosiphum euphorbiae* (Thomas) (Hemiptera: Aphididae) has been found to alter the rhizosphere microbiome of tomato plants. This alteration in the soil microbiome has a lasting impact, resulting in enhanced resistance to aphids in subsequent generations of these plants [20]. In summary, these studies suggest that insect herbivory can influence the dynamics of the plant rhizosphere microbiome and help suppress aboveground pests through plant–soil feedback mechanisms. However, the effects of insect herbivory on microbiome functions and microbial networks within the rhizosphere of host plants are still relatively underexplored. Deepening our understanding of the structure and function of plant-associated microbial communities under various environmental conditions could offer valuable insights for developing resource-efficient and stress-resistant agricultural ecosystems [21].

Tea is a significant economic crop globally, with over 60 tea-producing countries and regions generating nearly 6 million tons of tea annually. Notably, China had an impressive 3.43 million hectares of land dedicated to tea gardens in 2023 [22]. However, these tea gardens are frequently plagued by *Ectropis grisescens* Warren (Lepidoptera: Geometridae), a primary leaf-eating pest that is particularly prevalent in China’s major tea-producing areas, especially in the middle and lower reaches of the Yangtze River [23]. This pest, known as the grey geometrid moth, exhibits adaptability to a wide range of climatic conditions. With its high fecundity and short generation time, coupled with rapid dispersal rates, it presents a significant threat to tea production, leading to substantial economic losses [24]. Conventional management of *E. grisescens* has heavily relied on chemical pesticides, but their improper use can lead to pest resistance, pesticide residues, deteriorated tea quality, and pose public health risks [25,26]. Consequently, it is essential to urgently explore alternatives to chemical insecticides. One promising avenue is leveraging a plant’s inherent defense mechanisms against pests and diseases. Below-ground microorganisms can stimulate resistance in plants to aboveground insect pests. Studies indicate that plants inoculated with PGPR have latent defense mechanisms against pests and pathogens [27,28]. For example, fluorescent Pseudomonad strains can inhibit the development of *Cnaphalocrocis medinalis* (Guenée) (Lepidoptera: Pyralidae) (commonly known as rice leaf-folder) by inducing defense molecules in rice plants, which, in turn, enhance resistance to leaf-folder attacks. Furthermore, *Stenotrophomonas* sp. T6-4, which was isolated from the pest-infested tomato rhizosphere, has been found to enhance tomato growth and bolster tomato resistance against the insect herbivory of *Spodoptera litura* (Fabricius) (Lepidoptera: Noctuidae) [28]. These examples demonstrated that root-associated microorganisms could serve as a powerful defensive strategy against terrestrial insect attacks and should be integrated into pest management strategies [29]. To date, most studies have focused on using PGPR to manage plant pathogens, with limited literature exploring their role in controlling insect pests. Nevertheless, the induction of systemic resistance by PGPR against various pests and pathogens is widely regarded as the most promising approach in plant protection [30,31]. Consequently, deciphering the behavior of the root-associated microbiome in response to *E. grisescens* infestation could provide valuable insights for formulating more efficacious strategies to control this pest.

The interactions between aboveground insects and root-associated bacteria (rhizosphere bacteria), mediated by plants, are particularly intriguing. However, research on the intricate interspecies interactions among tea plants, aboveground insects, and rhizosphere microorganisms is limited. In this study, we designed a time-series experiment to explore the potential “cry for help” from tea plants in the tea-geometrids system. To achieve this, we employed 16S amplicon and metagenomic sequencing to analyze the root-associated microbiomes of both infested and non-infested plants, complemented by transcriptomic analyses and experimental assays. Our study aimed to fulfill three objectives: (i) to investigate the temporal variations in the composition and functionality of the root-associated microbiome between infested and non-infested plants; (ii) to identify key microbial members associated with geometrids infestation; and (iii) to uncover the functional pathways through which these key microbial members may enhance resistance to geometrid infestations.

## 2. Materials and Methods

### 2.1. Tea Plant Cultivation

A controlled greenhouse experiment was conducted using one-year-old *Camellia sinensis* cv. Huangguanyin tea seedlings. These seedlings were cultivated in plastic pots, each measuring 17 cm in height and 15 cm in diameter, with three healthy plants maintained per pot. The potted plants were then placed inside nylon mesh cages and grown under controlled conditions in a glasshouse set at a temperature of 25 ± 2 °C, with a lighting schedule of 16 h of illumination followed by 8 h of darkness.

### 2.2. Insects Rearing

*Ectropis grisescens*, obtained from the Key Laboratory of Biopesticide and Chemical Biology at Fujian Agriculture and Forestry University (Fuzhou, China), was reared on potted fresh tea shoots and maintained in an incubator under controlled conditions of 26 ± 2 °C temperature, 70 ± 5% relative humidity, and 16 h: 8 h of light: dark photoperiod. After one generation, the caterpillars were used for experimental purposes.

### 2.3. Insect Infestation

This study was conducted on 1 October 2023 in Wuyishan City, Fujian Province, China (latitude 27°7′ N, longitude 118°1′ E). Tea plants of uniform height and canopy width were selected for herbivory treatments. Third-instar *E. grisescens* larvae were subjected to an 8 h starvation period before being evenly distributed on the leaves at a density of ten larvae per tea plant. According to previous literature records [32], the optimal treatment time is around one-third of the leaf area for insects to feed on. The larvae were then allowed to feed for 6 h, taking 1/3 of the leaf area, after which they were removed. Control the environment variables of each processing group to be the same. A time-series experiment was designed (refer to Figure 1a), involving the collection of three biological replicates of tea seedling leaves, roots, and rhizosphere soil samples on the first (S1), second (S2), seventh (S3), and fifteenth (S4) days post-removal of *E. grisescens*. Control samples consisted of non-infested tea seedling leaves, root samples, and rhizosphere soil samples. All collected samples were immediately frozen in liquid nitrogen and then stored at −80 °C for further use.

### 2.4. Soil Samples 16S rRNA Gene Amplicon Sequence

Genomic DNA was extracted from 0.5 g of soil samples using the Soil DNA Isolation Kit from Omega Bio-tek (Norcross, GA, USA), following the manufacturer’s guidelines. To evaluate the purity and quantity of the isolated DNA, we pooled three distinct extracts from each sample and measured their absorbance at wavelengths of 260 nm and 280 nm using a fluorescence spectrophotometer (Quantifluor-ST fluorometer, Promega, E6090 (Madison, WI, USA); Quant-iT PicoGreen dsDNA Assay Kit, Invitrogen, P7589 (Carlsbad, CA, USA)). Additionally, we analyzed the integrity of the DNA via electrophoresis on a 1% agarose gel. Subsequently, the DNA solution was adjusted to the appropriate concentration and stored at 4 °C, while the stock solution was preserved at −20 °C.

The PCR amplification process commenced with an initial denaturation step at 95 °C for 2 min, followed by 20 cycles. Each cycle consisted of denaturation at 98 °C for 10 s, annealing at 62 °C for 30 s, and extension at 68 °C for an additional 30 s. A final extension was conducted at 68 °C for 10 min to complete the amplification. The products of the bacterial 16S rRNA gene amplification were visualized on a 2% agarose gel and then purified using the AxyPrep DNA Gel Extraction Kit (Axygen Biosciences, Union City, CA, USA). The concentration of the purified DNA was determined using the QuantiFluor-ST assay (Promega, Madison, WI, USA). The purified amplicons were combined in equimolar ratios and subjected to paired-end sequencing (2 × 250 bp) on the Illumina platform, following standard protocols. The sequencing library was prepared by Magigene services on the Illumina HiSeq2500 PE250 platform (Magigene Co., Ltd., Guangzhou, China).

### 2.5. Plant Samples RNA-Seq, De Novo Assembly, and Functional Annotation

Total RNA was extracted from the collected samples using the RNAprep Pure Plant Kit (Tiangen, China), following the manufacturer’s protocol. Equal amounts of RNA from samples taken at 1, 2, 7 and 15 days post-herbivory treatments were pooled for the EL (Leaves of the infested group), ER (Roots of the infested group), CKL (Leaves of the non-infected group), and CKR (Roots of the non-infected group) groups, with three biological replicates for each group. The quality of the extracted RNA was verified through gel electrophoresis and spectrophotometry using a Nanodrop instrument (Thermo Fisher in Waltham, MA, USA). This RNA was then utilized to prepare sequencing cDNA libraries, as outlined by Tai et al. [33]. Quality assessment of the sequencing libraries was performed using an Agilent 2100 bioanalyzer. Sequencing was conducted on the Illumina HiSeq 2500 platform by Aiji Baike Biotechnology Co., Ltd. (Wuhan, China). High-quality, clean reads were generated for each library after implementing quality control measures, and these reads were assembled using the Trinity short-read assembly program [34]. The assembled unigenes were annotated using BLASTx against various databases, including non-redundant protein (Nr) databases, the Swiss-Prot database, the Kyoto Encyclopedia of Genes and Genomes (KEGG) database, and the Cluster of Orthologous Groups (COG) database, with an E-value threshold set at 1 × 10^−5.34^ Gene Ontology (GO) classifications and Kyoto Encyclopedia of Genes and Genomes (KEGG) pathways were assigned to the assembled unigenes, following the methodology described by Shi et al. [34].

### 2.6. Plant Samples Identification of Differentially Expressed Genes (DEGs)

To identify the genes induced by *E. grisescens* infestation, we integrated all assembled unigenes from our transcriptomes with those from previously published transcriptomes related to this pest [32], as well as gene models derived from the tea plant genome [35]. These combined reference genes were then used to align the clean reads from each RNA-Seq dataset. The expression level of each gene was quantified using the FPKM method, implemented through Cufflinks (version 1.0.3) [36]. To identify differentially expressed genes (DEGs), we utilized the DEGseq R package [37] to contrast EL with CKL (designated as the local group) and ER with CKR (designated as the systemic group). We established a significance threshold for differential expression based on a false discovery rate (FDR) of ≤0.001 and an absolute log_2_ fold change of ≥1.

### 2.7. Co-Occurrence Network Analyses

To explore the inherent interactions within the microbial communities across the samples [38], we constructed microbial co-occurrence networks by analyzing the correlations in relative abundance at the genus level. A co-occurrence was deemed robust if it had a *p*-value below 0.05 and Spearman’s correlation coefficient (ρ) exceeding 0.70, indicating a statistically significant relationship between two genera. For visualizing these networks, we utilized Gephi 0.10.1 [39]. In the microbial network, nodes represent individual microbial genera, while edges illustrate biologically or biochemically meaningful connections between pairs of nodes. Using Gephi, we computed various network topological properties, such as the number of positive and negative edges, average path length, average degree, modularity, and average clustering coefficient. The significance of each node was evaluated based on its closeness centrality and its degree within the entire network [40], both of which were also calculated using Gephi. To ascertain statistical differences in the degree of the rhizosphere network among various groups, we applied the Kruskal–Wallis test.

### 2.8. Metagenomic Sequencing

DNA extracts from three infested rhizosphere samples and three non-infested rhizosphere samples were sequenced using pair-end sequencing on the Illumina NovaSeq platform. The raw sequence data were processed and trimmed using FastP [41]. To eliminate potential contamination from the host plant, the cleaned reads were aligned against the *C. sinensis* genome (accession: ASM 1731120v1) [42] using BWA [43]. The remaining metagenomic reads were assembled using MEGAHIT [44], and contigs shorter than 300 bp were discarded. To explore the impact of pest infestation on microbial functions in the tea rhizosphere, we supplemented our 16S sequencing data with metagenomic sequencing for a select group of samples. Following the sample procedure, we sequenced DNA extracts from five rhizosphere samples with pest infestation and five without, using pair-end sequencing on the Illumina NovaSeq system. The raw sequence data were processed and trimmed with FastP [41] to remove potential contamination from the host plant. Cleaned reads were then aligned to the genome of *C. sinensis* (accession: ASM 1731120v1) [42] using BWA [43]. The remaining metagenomic reads were assembled with MEGAHIT [44], and contigs shorter than 300 bp base pairs were discarded. Open reading frames from the assembled contigs were predicted using MetaGene [45] in metagenomics mode, and non-redundant gene sets were generated with CD-HIT [46], applying a similarity threshold of 95%. For taxonomic annotation, non-redundant genes were aligned to the NR database [47]. Functional annotations were conducted by aligning the genes to the KEGG database [48] using Diamond [49]. Lastly, we employed the Wilcoxon rank-sum test to assess variations in the relative abundance of KEGG Orthology (KO) terms.

### 2.9. Isolation and Characterization of Burkholderia

To isolate *Burkholderia* from the rhizosphere of tea plants, a 10 g soil sample was mixed with 90 milliliters of sterile water and shaken vigorously for 20 min. The resulting soil mixture was then poured onto Luria–Bertani (LB) agar plates and incubated at 30 °C for three days, with daily monitoring to identify and isolate the target colonies. Subsequently, the isolated bacteria underwent a purification process that included three rounds of subculturing. The 16S rRNA gene was amplified using the primer set 27F and 1492R. The amplified 16S rRNA gene sequences were subsequently deposited in the NCBI database (accession number available in GSA: Appendix A) (https://www.ncbi.nlm.nih.gov/, accessed on 25 May 2024). For phylogenetic analysis, we employed MEGA 11 software to construct a phylogenetic tree using the neighbor-joining method, with bootstrap analysis conducted to assess the tree’s reliability. Ultimately, this methodology facilitated the isolation of nine unique *Burkholderia* strains.

### 2.10. Effects of Burkholderia Isolations on Plant Growth

Nine *Burkholderia* strains were cultured in 18 mm × 180 mm glass tubes at 32 °C with shaking at 200 rpm in 10 mL of LB for two days. Following centrifugation, each strain was rapidly washed three times with sterile water. The optical density (OD_600_) of each strain was adjusted to between 0.6 and 0.8 using sterile water.

To determine which of the nine isolated single bacteria positively influenced plant growth, tomato seeds were employed as experimental subjects. The seeds underwent surface sterilization in a 3% NaClO solution for 1 min, repeated thrice, and were subsequently rinsed with sterile water three times. Seeds exhibiting consistent size were selected and immersed in suspensions of nine different bacterial strains at 26 °C for 2 h, then placed on sterile filter paper saturated with sterile water within a 100 mm × 100 mm Petri dish under conditions of 26 °C and 12 h of light to facilitate germination. Each Petri dish contained fifteen seeds, and there were three biological replicates per group. After five days, the length, fresh weight, and dry weight of the seedlings were measured as an indicator of seedling growth. Consequently, three bacterial strains—*Burkholderia cepacia* strain ABC4, *Burkholderia* sp. strain SSG, and *Burkholderia cepacia* strain B-6—which exhibited superior growth-promoting effects, were selected for further inoculation verification experiments and labeled as T1, T2, and T3, respectively. Additionally, a synthetic community1 (T4) was constructed using these three strains mixed in equal proportions, and a synthetic community2 (T5) using nine strains mixed in equal proportions. The control group (CK) was treated with sterile water.

In accordance with the experimental design, five treatments labeled T1 through T5 were prepared as 200 mL bacterial suspensions. Each treatment group consisted of ten pots of tea seedlings, for a total of sixty pots used in the study. Each pot of tea seedlings was inoculated with 10 mL of the bacterial suspension twice per week for one month. Subsequently, the plants were harvested, and the fresh root weight and fresh shoot weight were determined.

### 2.11. Larvae Feeding Selection Bioassay

Under six different treatment conditions, leaves of the same position and size were collected, and their area was calculated using ImageJ2 software (version2.14.0) before the feeding experiment. Subsequently, six leaves, one from each of the six different treatments, were arranged randomly on a disk. Six third-instar larvae, which had been starved for 8 h, were then placed in the center of the disk. After a feeding period of 3 h, the larvae were removed, and the remaining leaf area was measured to determine the percentage of the leaf that had been consumed.

### 2.12. Chemical Defense Substances Analysis

The uppermost leaf was defined as young, and the first two leaves were defined as old; the leaves situated between these two extremes were considered as ’intermediate’ and selected for sampling. For each sample, weighing 100 mg, the leaves were blended with 900 μL of phosphate-buffered saline (PBS) and then centrifuged at 5000 rpm for 15 min. The supernatant obtained was meticulously collected for further analysis of plant trypsin inhibitors, jasmonic acid levels, superoxide dismutase (SOD) enzyme activity, and peroxidase (POD) enzyme activity. The analysis of plant trypsin inhibitor was conducted using the plant trypsin inhibitor ELISA kit (Zoman Bio-tek, Nanjing, China), while jasmonic acid levels were measured with the plant JA ELISA Detection Kit (Zoman Bio-tek, Nanjing, China). The activities of SOD and POD were measured using the enzyme activity test kit (Sino Best Biotech, Shanghai, China).

Furthermore, the intermediate leaves with different treatments were dried until the quality of the tea sample remained unchanged, then ground into tea powder. The polyphenol content of the tea powder was analyzed using the Folin phenol colorimetric method [50]. The determination of flavonoids was performed using the aluminum chloride colorimetric method, while caffeine levels were measured using an ultraviolet (UV) spectrophotometer [51,52].

### 2.13. Statistical Analysis

Alpha and beta diversity metrics were assessed using the QIIME software (version 1.7.0). Visualization of the principal coordinate analysis (PCoA) was conducted using the FactoMineR and ggplot2 packages in R software (Version 2.15.3). Real-time data analysis was conducted with SPSS version 19.0 (SPSS, Inc., Chicago, IL, USA). To compare plant traits and chemical defense substances among the treatments, a one-way analysis of variance (ANOVA) was conducted, followed by Tukey’s honestly significant difference (HSD) test.

## 3. Results

### 3.1. Ectropis grisescens Infestation Restructures the Tea Rhizosphere Microbiome Assembly

To better understand the dynamic changes in the rhizosphere soil induced by *E. grisescens* infestation, a time-series experiment was conducted. Rhizosphere soil was collected from the treatment group at each time point (S1, S2, S3, S4) and compared with that of tea seedlings non-infected by *E. grisescens* (Figure 1a). Four contrasting groups were established: E1 vs. CK1, E2 vs. CK2, E3 vs. CK3, and E4 vs. CK4, for expression comparative analysis at corresponding time points. The rhizosphere soil was analyzed using 16S rRNA amplicon sequencing, and after filtering the raw sequencing data, high-quality CCS reads from the sequencing platform were retained. The rarefaction curve for each group suggested that bacterial diversity curves nearly reached the asymptote, indicating sufficient sequencing depth for most samples (Figure 1b).

On average, there were 103,458 soil bacterial community sequences per sample, and comparative diversity analyses were performed after normalizing sequence numbers to the lowest sequencing depths (5267 reads per sample). An alpha diversity analysis of these eight groups of rhizosphere bacterial communities was conducted using the Chao1 index for richness and the Shannon index for diversity. A box plot based on the Chao1 index and Shannon index showed no significant difference between these two indicators and the control group at S1 and S2, but they were significantly higher than the control group at S3 and S4 (Figure 1c). PCoA analysis (Figure 1d) also found that at times S3 and S4, the larvae-infested group and the control group could be distinctly separated (PERMANOVA; *p* < 0.05), whereas the larvae-infested group at S1 and S2 could not be clearly differentiated from the control group. Both findings suggest that the first two days post-larvae infestations have minimal impact on the bacterial community structure in the rhizosphere soil of tea seedlings.

Bacterial OTUs were classified into 35 known bacterial phyla, 91 classes, 167 orders, 214 families, and 329 genera. Notably, Acidobacteria, Proteobacteria, Actinobacteria, Bacteroidetes, Chloroflexi, Patescibacteria, Planctomycetes, Proteobacteria, WPS-2, and Verrucomicrobia were the predominant phyla. A cumulative bar chart of relative abundance at the phylum level indicated no significant differences among these groups (Figure 1e).

Next, we explored the disparities in the root microbiota between the larvae-infected group and the non-infected group at four time points (S1, S2, S3, S4) at the genus level. At these time points, genes enriched in rhizosphere soil spanned a wide array of bacterial phyla, including Acidobacteria, Proteobacteria, Actinobacteria, Bacteroidetes, Chloroflexi, Firmicutes, Patescibacteria, Spirochaetes, and Verrucomicrobia (FDR adjusted *p* < 0.05, Wilcoxon rank sum test) (Figure 2a). The Manhattan diagram illustrated that the size of the graph represented the magnitude of the Fold Change (Figure 2a). Through this diagram, we discerned significant differences (ANOVA, Tukey’s HSD, *p* < 0.05) in bacterial genera at these four time points, primarily within the five phyla: Actinobacteria, Bacteroides, Firmicutes, Proteobacteria, and Verrucomimicrobia. Although notable differences in bacterial genera were observed at times S1 and S2, their fold changes were relatively minor. In contrast, at times S3 and S4, a greater number of significantly upregulated bacterial genera emerged in the group infested by *E. grisescens* larvae, presenting a stark contrast to times S1 and S2. This further suggested that the soil rhizosphere bacterial community, post-infested by *E. grisescens,* cannot rapidly reassemble and necessitates an extended period for recovery.

A comparative analysis at the genus level was conducted between the two groups affected by *E. grisescens* infestation at times S3 and S4. The volcano plot visualized the differential bacterial genera, revealing that only seven genera exhibited significant differences with fold changes greater than 2 (Figure 2b). Moreover, the plot’s dot sizes represent the relative abundance of these bacterial genera, revealing that their collective abundance in the samples was minimal. This indicated that there was no substantial difference between the two groups affected by *E. grisescens* larvae infesting at times S3 and S4. It also suggested that by the seventh day following *E. grisescens*’s infesting (time S3), the rhizosphere soil bacterial community had already reassembled and begun to stabilize.

Therefore, our research on rhizosphere soil concentrated on time point S3. Initially, we filtered out genera with a relative abundance below 0.001. Subsequently, we selected genera with a relative abundance exceeding 0.01 and those displaying a differential fold change greater than 2 (|logFC| > 1) for an in-depth analysis, which was presented in the form of a bar graph (Figure 2c). This approach allowed us to observe the differences between the group affected by *E. grisescens* and the non-infected group at time S3. The bacterial genera that exhibited a significant increase in relative abundance > 0.01 included *Bradyrhizobium*, *Burkholderia-Caballeronia-Paraburkholderia*, *Sphingomonas*, *Mucilaginibacter*, *Candidatus* Solibacter, and *Candidatus* Udaeobacter. In addition, *Chthoniobacter*, *Dyella*, *Niatella*, *Massilia*, *Ktedonobacter*, *Sphingobium*, *Bdellovibrio*, *Pedomicrobium*, *Oryzihumus*, *Ramlibactor*, and *Bacillus* significantly increased among bacterial genera with a difference multiple > 2.

### 3.2. Ectropis grisescens Infestation Triggers Transcriptomic Reprogramming in Tea Plants

In this study, we extended our prior time-series experiments using transcriptome profiling data to investigate the gene expression alterations in tea plants after *E. grisescens* infestation, aiming to correlate these changes with the alterations in rhizosphere bacterial community. We conducted transcriptome sequencing on leaves and roots from both larvae-infested and non-infected plants at four time points. At each time point, the two groups served as natural comparison pairs (S1:E1 vs. CK1, S2:E2 vs. CK2, S3:E3 vs. CK3, S4:E4 vs. CK4). A principal component analysis of gene expression initially verified the reliability and appropriateness of the experimental setup, as different time points were clearly separated into distinct clusters, with minimal variations between the biological replicates at each time (Figure 3a). Applying significance criteria of fold change ≥ 2, BH-adjusted *p* < 0.05, and VIP > 1 (Variable importance in the projection), we identified DEGs across various groups and created multiple differential gene volcano maps (Figure 3b). These plots revealed that the highest number of differentially expressed genes in both leaves and roots occurred in the S2 group.

We performed a KEGG enrichment pathway analysis on the differentially expressed genes (DEGs) identified in the leaf and root tissues of the S2 group. This analysis revealed that the DEGs in the leaf tissue were enriched in 42 pathways, while those in the root tissue were enriched in 32 pathways, with an overlap of 18 common pathways between them (Figure 3c). However, a subsequent statistical analysis of the genes involved in these pathways demonstrated that the root tissue exhibited a higher number of differentially expressed genes than the leaf tissue within the common pathways shared by both groups. These findings suggest that, compared to insect-damaged leaves, *E. grisescens* herbivory induces a larger scale of transcriptional rearrangement in specific pathways within undamaged roots underground. Notable examples include pathways related to phenylpropanoid biosynthesis, galactose metabolism, and plant hormone signal transduction.

Our research is specifically focused on the underground root system; consequently, we constructed a pathway regulatory network for the pathways enriched in this system (Figure 3d). This network enables us to develop deeper into the core metabolic pathways occurring within the root system. Utilizing this network, we identified several significantly altered metabolic pathways. These include the plant signal transduction pathway—plant hormone signal transduction, and two secondary metabolic pathways: alpha-linolenic acid metabolism and phenylpropanoid biosynthesis. Notably, we observed a strong link between the phenylpropanoid biosynthesis pathway and another node, the flavonoid biosynthesis pathway. The latter is a downstream branch of the phenylalanine metabolic pathway. These two pathways share numerous regulatory genes, drawing our attention to the potential importance of the flavonoid biosynthesis metabolic pathway.

Based on these three secondary metabolic pathways, we generated a differential gene expression map of metabolic pathways (Figure 4), providing a visual representation of the transcriptional changes occurring within these pathways in response to *E. grisescens* larvae infestation. This map allowed us to pinpoint specific genes and pathways that are affected by the infestation. Utilizing the established biosynthesis pathways of phenylpropanoids and flavonoids in model plants as references, we formulated a regulatory network diagram of DEGs in tea trees (Figure 4). Our analysis of this network revealed that a significant number of genes involved in the biosynthesis of phenolic acid compounds, flavonoids, caffeine, and alpha-linolenic acid were activated in response to the larval infestation.

### 3.3. Ectropis grisescens Infestation Destabilizes Rhizosphere Microbiota Co-Occurrence Networks

To further understand the impact of *E. grisescens* infestation on the rhizospheric bacterial community of tea plants, a co-occurrence network analysis was conducted at the S3 time point, focusing on bacterial genera. Significant correlations were identified through Spearman’s correlation test (*p* < 0.05, r > 0.7), leading to the construction of co-occurrence networks for both larvae-infested and non-infested groups.

Both networks comprised an equal number of nodes, totaling 105. However, the larvae-infested group exhibited fewer edges (1425) compared to the non-infested group (1608), marking a reduction of 183 edges. This reduction suggested that infestation by the *E. grisescens* caterpillar disrupted the stability of the rhizospheric microbial network (Figure 5a). Within this network, the number of modules increased from three to four. The average clustering coefficient rose from 0.805 to 0.839, while the average path length increased from 1.567 to 1.643. Despite the overall decrease in edges from 1608 to 1425, the ratio of positive to negative edges maintained a consistent 2:1 ratio (Figure 5b). Moreover, the mean degree centrality experienced a decline in the infested rhizosphere microbiota network relative to its non-infested counterpart (Figure 5c; Kruskal–Wallis test, *p* < 0.05).

To pinpoint the most influential microbes within these networks, we zeroed in on “hubs”—microorganisms with high connectivity in the scale-free correlation networks [40]. We computed topological features such as degree, betweenness centrality (Figure 5d), and closeness centrality (Figure 5e) for individual nodes. Intersecting the significant classification units from these three connectivity parameters revealed 10 bacterial genera exhibiting elevated closeness centrality and betweenness centrality: *Burkholderia*, *Edaphobacter*, *Trinickia*, *Methylobacterium*, *Methylovirgula*, *Novosphingobium*, *Mycobacterium*, *Pseudomonas*, *Sphingobium*, and *Arthrobacter*. These genera demonstrated higher node degrees in the larvae-infested group, establishing them as key hubs in the entire rhizospheric microbial network. Notably, *Burkholderia* [53] and *Sphingobium* [54] have been associated with nitrogen metabolism.

### 3.4. Ectropis grisescens Infestation Affects Tea Rhizosphere Microbiome Function

To further investigate the functional alterations in the rhizosphere microbiome induced by *E. grisescens* infestation, we conducted metagenomic sequencing on rhizosphere samples from both infested and non-infested plants. This process yielded a total of 439,693,642 raw reads from the samples. Due to the presence of a large number of low-quality bases, adapter contamination, or short lengths in plant-derived sequences, these factors can affect the accuracy of subsequent analysis. Therefore, quality checks and filtering were performed on the sequence, resulting in a total of 348,848,776 clean readings. From the assembled sequences, 2,854,967 potential protein-coding genes were predicted. These genes were then clustered at a 90% identity threshold, resulting in a final compilation of 2,443,384 non-redundant genes.

To assess the impact of *E. grisescens* infestation on the functional characteristics of the microbiome, the non-redundant genes were annotated using the KEGG database, which resulted in the identification of 6253 KEGG Orthologs (KOs). A differential enrichment analysis was conducted between the infested and non-infested plants, revealing 115 enriched KOs and 167 depleted KOs (Figure 6a; Wilcoxon rank-sum test, *p* < 0.05). It is worth noting that the majority of enriched KOs are associated with metabolic pathways that reflect the adaptation and changes made by soil microorganisms in tea plants under pest stress (Figure 6b).

Specifically, the enriched KOs were involved in pathways of carbohydrate metabolism (29 KOs), amino acid metabolism (23 KOs), energy metabolism (16 KOs), lipid metabolism (4 KOs), nucleotide metabolism (12 KOs), metabolism of terpenoids and polyketides (10 KOs), metabolism of cofactors and vitamins (8 KOs), xenobiotics biodegradation and metabolism (9 KOs), biosynthesis of other secondary metabolites (1 KO), and metabolism of other amino acids (2 KO) Appendix A. These identified metabolic pathways play important roles in regulating plant growth and defense. Carbohydrate metabolism provides energy and a carbon backbone for plant growth [55], amino acid metabolism regulates plant defense signals and antioxidant systems [56,57], and nitrogen metabolism affects the synthesis and distribution of plant defense substances [58,59]. The synergistic effects of various metabolic processes in plants form an energy defense dynamic balance, jointly enhancing their insect resistance and stress resistance.

During the nitrogen metabolism pathways (ko00910), multiple genes involved in nitrogen fixation, nitrification, and denitrification were found to be enriched. Among them, the relative expression levels of NifD, AmoA, Hao, NirB, NirD, and NrfA in the rhizosphere microbiome of infested plants were higher than those in non-infested plants (Figure 6c). The upregulation of these genes not only indicates an intensification of nitrogen transformation in the soil but also suggests an increase in the available nitrogen, specifically ammonium content, in the soil. And in our subsequent experiments, we measured the transcriptome of plant roots and found that the nitrogen metabolism pathway in roots was significantly upregulated (not yet published), indicating that upregulation of these genes can improve plant nitrogen absorption and transformation.

To investigate the relationship between community members and gene functionalities, we employed a Circos plot to analyze the contribution of various community members to genes associated with the nitrogen cycle (Figure 6d). We observed that *Burkholderia*, *Bradyrhizobium*, and *Novosphingobium* were the major contributors to those genes (Figure 6d). The results indicate that *Burkholderia*, *Bradyrhizobium*, and *Novosphingobium* in the rhizosphere may have a significant role in the ammonification process, potentially enhancing the plant’s response to the aboveground infestation by *E. grisescens*. Among these genera, *Burkholderia* was found to be involved with the greatest number of genes and accounted for the highest proportion, indicating its potential as the most crucial bacterial genus in the response to pest stress.

Subsequently, we conducted a more detailed analysis of the *Burkholderia* genus, and the Sankey plot showed the 13 annotated bacterial species (Figure 6e). The width of the Sankey line represents the percentage of distribution. Among them, *Burkholderiaceae_bacterium* and *Burkholderiales_bacterium* have the highest proportion. The expression differences in these strains among different treatments were analyzed through heatmaps, and it was found that the expression levels of *Burkholderia_cenococcia* and *Burkholderia_cepacia* in the group infested by the *E. grisescens* were higher than those in the non-infested group.

### 3.5. Burkholderia Positively Enhance Plant Pest-Resistant Performance

To elucidate the function of *Burkholderia* in a plant–insect interaction, nine distinct *Burkholderia* strains were isolated from the rhizosphere (Figure 7b). Their growth-promoting capabilities were confirmed via a flat plate growth promotion experiment (Figure 7a), with detailed results provided in the Appendix A. Five different treatment combinations (T1–T5) were constructed using nine individual bacterial strains, with a control group (CK) inoculated with sterile water. The evaluation of shoot and root fresh weight was conducted across six treatments: CK, T1, T2, T3, T4, and T5. Notably, T1, T2, and T5 treatments significantly increased both root fresh weight and shoot fresh weight of tea seedlings, thereby fostering their growth (both *p* < 0.05) (Figure 7d,e).

In the subsequent experiment, we conducted a selective feeding trial on the *E. grisescens* and analyzed its feeding preference by calculating the percentage of leaf area consumed over time (Figure 8a). The results revealed that only the T1 treatment group had a significantly smaller larvae-consumed leaf area compared to the control group (Figure 8b, ANOVA, Tukey’s HSD, *p* < 0.05).

For our bacterial feedback validation experiment, we focused on studying the expression levels of certain secondary metabolites commonly used as insect-resistant substances in tea plants. We verified the content of polyphenols, flavonoids, caffeine, jasmonic acid, and other substances among the groups. The results showed that the content of flavonoids and polyphenols in the T1 treatment group was significantly higher than that in the control group (Figure 8c,e, ANOVA, Tukey’s HSD, *p* < 0.05), while the content of caffeine and jasmonic acid was significantly higher than that in the control group (Figure 8d,f, ANOVA, Tukey’s HSD, *p* < 0.01). The T5 group also had a significantly higher content of caffeine and jasmonic acid compared to the control group (ANOVA, Tukey’s HSD, *p* < 0.01). However, the treatment of other groups (T2, T3, T4) had no significant effect on these indicators, and even led to a decrease in the content of flavonoids and polyphenols in tea leaves under T3 and T4 treatment.

Additionally, we observed the content of defense enzymes and trypsin inhibitors in each treatment group. The two defense enzymes (SOD, POD) in the T1 and T5 groups were significantly higher than in the control group (Figure 8g,h, ANOVA, Tukey’s HSD, *p* < 0.01), while the SOD activity in the T4 group was significantly lower than in the control group (ANOVA, Tukey’s HSD, *p* < 0.05). Meanwhile, the other groups had no significant effect on these two indicators. In terms of trypsin inhibitor content, the levels in both T1 and T5 groups were significantly higher than those in the control group (Figure 8i, ANOVA, Tukey’s HSD, *p* < 0.01), and the levels in T2 group were significantly higher than those in the control group (ANOVA, Tukey’s HSD, *p* < 0.05).

## 4. Discussion

Rhizosphere microorganisms, often referred to as the “second genome” of plants, are pivotal in enhancing their resilience against stress. Research indicates that plants employ a “cry for help” strategy to counteract various biotic and abiotic challenges [60]. When under pest-induced stress, the plants not only synthesize defensive compounds but also release “cry for help” signals to attract beneficial organisms, such as natural enemies and soil microorganisms, to help mitigate the stress. Additionally, the roots of plants employ a similar strategy by secreting metabolites that recruit helpful soil microbes [4,6,8]. Research has documented the effects of insects on plant root microbiota and the reciprocal feedback effects of microbiota on plants across various species, including cowpea [8], tomato [20], and cabbage [61], etc. Tea, as an economically significant crop cherished globally, is particularly vulnerable to pest infestations during its cultivation, resulting in diminished yield and quality [62]. Despite the scarcity of studies investigating the complex interactions among insects, plants, and root microbiota specifically in tea trees, it remains unclear how the rhizosphere microbiota of tea trees responds to insect pest stress. This study aims to elucidate the alterations and functional significance of root-associated microbial communities under herbivory stress, using *E. grisescens* and the tea plant as the model system.

To analyze the temporal dynamic changes in the rhizosphere bacterial community of tea seedlings under herbivory stress by tea geometrids, we conducted a time-series experiment. The results revealed that within the initial two days post-pest stress, minor alterations were observed in the bacterial community of the rhizosphere soil. However, by time S3, a marked difference emerged between the rhizosphere soil subjected to pest stress and the control soil without pest stress. Thus, in response to the damage inflicted by aboveground herbivorous insects, the alteration observed in the rhizosphere bacterial community of tea trees indicates a potential delay. Delayed adjustments in bacterial communities have been observed in several studies. For instance, whitefly infestation reshaped the rhizosphere microbiota structure of pepper within a week, and increased the abundance of proteobacterial groups after two weeks [19]. In tomato plants grown in three different soil environments, significant changes in the rhizobacterial communities were only evident on the seventh day following aphid infestation [20]. In addition, a comparison of the rhizosphere soil samples at times S3 and S4 showed no substantial difference, indicating that by the 7th day after *E. grisescens*’s infestation (time S3), the bacterial community in the rhizosphere soil had already reassembled and begun to stabilize.

Further analysis of the genera that showed significant differences at the S3 time point revealed a noteworthy pattern. We found that several genera with significantly increased relative abundances—*Bradyrhizobium*, *Burkholderia*, *Sphingomonas*, *Mucilaginibacter,* and *Bacillus*—are well documented as rhizosphere-promoting bacteria, known not only to promote plant growth but also to assist plants in enhancing their resistance to biotic stress [63,64,65]. Therefore, the increase in these genera may play a crucial role in helping tea seedlings withstand the stress induced by *E. grisescens* attack. This suggests that the shifts in the rhizosphere bacterial community following herbivore damage could be instrumental in fostering the resilience of tea trees against such biotic challenges.

Herbivory by insects significantly impairs leaf photosynthesis, leading to a nutrient imbalance [66]. This imbalance may exert selective pressure on the root microbiome of tea seedlings, resulting in the restructuring of the rhizosphere microbial community [8]. To ascertain whether alterations in soil bacterial communities stem from herbivore stress imposing selective pressure on the root microbiome of tea seedlings, we delved into the transcriptional changes in tea seedlings affected by the *E. grisescens* infestation. Compared to insect-damaged leaves, those subjected to herbivory initiate more substantial transcriptional rearrangements in specific pathways within undamaged roots. Our findings revealed that numerous genes linked to the biosynthesis of phenylpropanoids, flavonoids, phenolic acid, and alpha-linolenic acid in the root system were activated. Existing studies have elucidated that these substances were instrumental in attracting nitrogen-fixing bacteria, such as *Burkholderia*, *Bradyrhizobium*, and *Sphingomonas*, which were significantly enriched at the S3 time point in our study [67,68,69]. Lin et al. found that catechin, a flavonoid compound, was highly expressed in the rhizosphere of *Pteris vittata* L. when induced by arsenic, and that *Bradyrhizobium* exhibited strong chemotactic behavior toward it [67]. Hartwig et al. found that the swelling effect of alfalfa seeds catalyzed the accumulation of quercetin-3-O-galactoside (a flavonol) and luteolin-7-O-glucoside (a flavonoid), both compounds shown to augment the growth rate of *Rhizobium meliloti* [68]. The phenomenon of flavonoids being highly expressed in root exudates, induced by autotoxic ginsenoside stress, facilitated the recruitment of the beneficial *Burkholderia* B36 strain [69]. In our study, crucial genes responsible for cinnamic acid synthesis, including six PAL genes, exhibited heightened expression. This could be a significant factor contributing to the notable rise in the abundance of the *Burkholderia* and *Bradyrhizobium* genera at time S3. Furthermore, the alpha-linolenic acid metabolism pathway in the roots of tea seedlings, post-infestation by the *E. grisescens,* became markedly activated. This activation resulted in an accumulation of linolenic acid and linoleic acid. Li et al. have also reported that the representative strain, *Sphingomonas* sp., was drawn to the surplus root exudates of linolenic acid and linoleic acid produced following pesticide application [70]. Perhaps the increase in the relative abundance of *Sphingomonas* is associated with the elevated expression of genes related to linolenic acid and linoleic acid. In summary, the microorganisms attracted to the roots of tea plants after stress induced by *E. grisescens* were correlated with the root exudates discharged by the tea plants. These exudates were primarily composed of substances such as flavonoids, phenolic acids, and fatty acids. Research has indicated that the microorganisms recruited by root exudates may serve as key members within the root microbial network [71,72].

The root-associated microbiomes constitute intricate, structured, and interconnected microbial networks [40,73], at the heart of which lie core microorganisms that act as hubs. These highly connected microbial species wield significant influence over the microbial community [40]. Notably, the presence of a microbial hub can modulate the responses of both plants and their microbiomes to dynamic environmental conditions [74]. Among the factors shaping this belowground ecosystem, root exudates play a crucial role. They can directly affect hub microbes, which, in turn, relay these environmental cues to the broader microbial community through intricate microbe–microbe interactions [40]. Hence, substances synthesized by genes with heightened expression in the transcriptome are highly likely to lure central microorganisms. With genera such as *Burkholderia*, *Bradyrhizobium*, and *Sphingomonas* that can be drawn to these secretions, it is reasonable to speculate that they might be core members of this microbial network.

To validate our hypothesis, we engineered a microbial symbiotic network. Given that the core microorganisms function as the linchpins of the network, identifying these key players is essential for deciphering the defense mechanisms of tea trees against geometrid damage. We computed the topological attributes for each node and found that the overlap of significant classification units from three connectivity parameters identified 10 bacterial genera with elevated closeness and betweenness centralities. These genera included *Burkholderia*, *Pseudomonas*, *Sphingobium*, *Edaphobacter*, *Trinickia*, *Methylobacterium*, *Methylovirgula*, *Novosphingobium*, *Mycobacterium*, and *Arthrobacter*. These are the key microorganisms within the rhizosphere bacterial network following the infestation by *E. grisescens*. This finding largely aligns with our previously proposed hypothesis. Notably, taxa such as *Burkholderia* [75,76,77,78], *Sphingomonas* [79,80], *Methylobacterium* [81,82,83], and *Pseudomonas* [84] are recognized for their roles in nitrogen fixing, nitrification, and denitrification, processes integral to soil nitrogen cycling. Consequently, our findings indicate that plants preferentially associate with microbes involved in nitrogen metabolism as key nodes in the rhizosphere when challenged by insect herbivory.

Furthermore, the results of metagenomic sequencing corroborated this finding. To investigate the relationship between community members and gene functions, we used a Circos plot to analyze the functional contributions of community members to the upregulated expressed genes involved in the soil nitrogen cycling process. We observed that, compared to the control group, the genera *Bradyrhizobium*, *Burkholderia*, and *Sphingomonas*, which showed a significant increase in relative abundance, were the primary contributors to these genes. *Burkholderia*, *Bradyrhizobium,* and *Novosphingobium* are all soil nitrogen cycling-related microorganisms that contribute many related genes in the process of soil nitrogen cycling. *Burkholderia* [75,85] and *Bradyrhizobium* [86,87] convert nitrogen in the air into ammonia through nitrogen fixation, providing important nitrogen sources for plants and promoting their growth and development. In addition, *Novoschingbium* [88] participate in ammonification, decomposing organic nitrogen compounds into ammonia, providing additional nitrogen sources for plants, and promoting nitrogen cycling. Especially *Burkholderia*, which involves the most genes and accounts for the largest proportion, may be the most important genus of bacteria involved in pest stress response. Additionally, it is a core microorganism within the bacterial interaction network and has strong connections with other bacterial genera in the network. Therefore, we hypothesize that *Burkholderia* plays a crucial role in assisting plants to resist pest stress.

Numerous studies indicated that root-associated microbial communities can alleviate nutritional deficiency stress in plants and enhance nitrogen uptake under deficient conditions [89,90]. Furthermore, the supplemental nitrogen furnished by these microbes can be transported to the stem, facilitating the synthesis of nitrogenous defensive compounds such as glucosinolates, cyanogenic glycosides, alkaloids, diverse peptides, proteins, and non-protein amino acids, thereby aiding in herbivore regulation [91,92,93]. Previous research on *Burkholderia*’s role in biological control predominantly emphasized its defense against and remediation of plant pathogens, with limited exploration into insect resistance. Therefore, the question arises: does the recruitment of *Burkholderia* at the damaged roots of tea seedlings serve a nitrogen-supportive role, enabling the seedlings to produce defensive agents against pest while ensuring unimpeded growth? Additionally, is the enhanced insect resistance in plants attributed to a singular strain or a collective microbial consortium?

To address these questions, we successfully cultivated nine distinct *Burkholderia* strains and selected three individual strains and two mixed cultures for reinoculation into the tea seedling. Our research found that the leaf area consumed by *E. grisescens* was significantly less in the five inoculation treatment groups compared to the control group, with the T1 treatment group (*Burkholderia cepacia* strain ABC4) showing a more substantial difference. Further analysis revealed an increase in the concentration of defensive nitrogen compounds such as flavonoids, polyphenols, and trypsin inhibitor in the aboveground parts of the treated tea seedlings. An elevation in nitrogen metabolism levels serves as a potential pathway to mitigate damage from external stressors [94,95]. Protease inhibitors, crucial for natural plant defense, were notably increased in the tea seedlings under T1, T2 (*Burkholderia* sp. strain SSG), and T5 (a synthetic community using nine *Burkholderia* strains) treatments. Many studies have revealed the adverse effects of trypsin inhibitors on pests [96,97]. Trypsin inhibitors bind to trypsin in the digestive tract of insects to form enzyme inhibitor complexes (EI), which block or weaken the hydrolysis of food proteins by trypsin, leading to obstruction of insect nutrient absorption and interference with their growth and development, and directly enhancing their pest resistance. In addition, the increased polyphenols, flavonoids, and caffeine, as constitutive defense compounds in plants, affect the feeding, growth, and survival of herbivores [98]. These substances have a strong bitter taste, which can affect the oral muscle activity of herbivores and reduce the feeding efficiency of pests [99]. Polyphenols can bind to digestive enzymes in the digestive tract of pests, reducing their digestive capacity and affecting their growth and development [100]. Caffeine has neurotoxicity to insects, which can affect their behavior and survival ability [99]. Considering these nitrogen-containing compounds, the T1 and T5 treatment groups demonstrated the most effective pest resistance. We also monitored the levels of protective enzymes in the tea seedlings. Superoxide dismutase (SOD) and peroxidase (POD), key protective enzymes regulated by the reactive oxygen species (ROS) that are essential for a plant’s response to adverse stressors [101], were significantly increased in the T1 and T5 groups, enhancing the levels of leaf protective enzymes. In summary, our study concluded that a single bacterial treatment (T1) and a mixed strain treatment (T5) significantly enhance the pest resistance of the plants. Especially, the T1 treatment group showed the most significant results. We conducted feeding experiments with *E. grisescens*, and the results showed that the leaves of tea seedlings treated with T1 were consumed the least by the *E. grisescens*. Subsequently, the expression levels of insect-resistant compounds in tea seedling leaves were measured, and the results showed that the levels of insect-resistant compounds were higher in the T1 treatment. Therefore, the experimental results of the feeding preference of the *E. grisescens* are consistent with the expression levels of insect-resistant compounds in tea seedling leaves and also reveal that the feeding preference of the *E. grisescens* is mainly influenced by different bacterial treatments on plant chemical characteristics. This further indicates that T1 treatment enhances the level of insect-resistant compounds in tea seedling leaves after inoculation with bacteria, thereby affecting the feeding behavior of the *E. grisescens*. Therefore, we suggested that the *Burkholderia* species recruited by the roots of tea seedlings affected by the infestation of *E. grisescens* could provide nitrogen support, synthesize nitrogen-based defensive compounds, and help improve the pest resistance performance of the tea seedlings.

## 5. Conclusions

Our research demonstrated that the rhizobacterial community of tea seedlings, when subjected to *E. grisescens* attack, experienced significant changes. In response, the rhizosphere mobilizes nitrogen (N) metabolism-related microbes to boost plant growth and enhance defense against *E. grisescens*. Notably, nitrogen-fixing bacteria *Burkholderia* species were actively recruited and became central players within the rhizosphere. When isolated strains of *Burkholderia* were reintroduced to tea plants, they significantly enhanced the tea plant’s resistance to insect pests, underscoring their potential as biocontrol agents. This study emphasizes the potential synergy among the rhizosphere microbiome, herbivores, and host plant, suggesting opportunities for innovative strategies to manipulate the rhizosphere microbiome to enhance insect resistance, improve crop yield, and promote sustainability.

## Figures and Tables

**Figure 1 insects-16-00412-f001:**
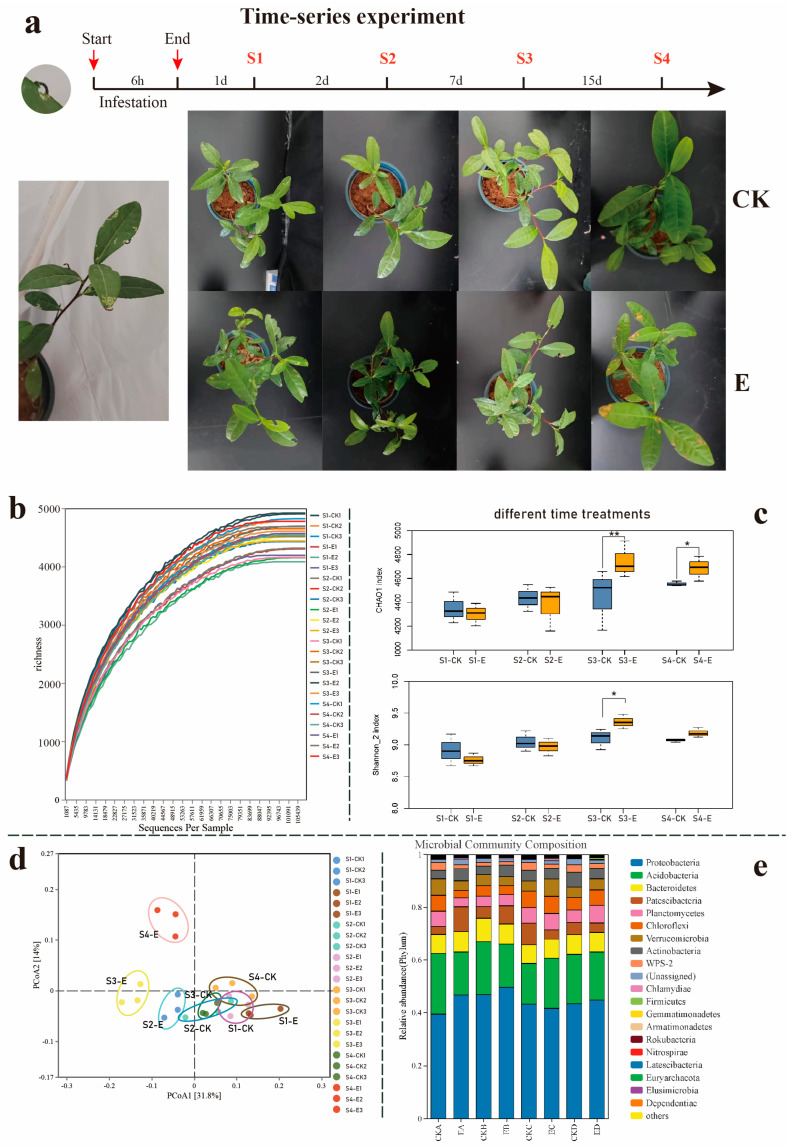
(**a**) Experimental design: Tea plants were exposed to *E. grisescens* for one day in a controlled glasshouse environment. Subsequently, leaf, root, and rhizosphere soil samples were collected at 1 (S1), 2 (S2), 7 (S3), and 15 (S4) days after the removal of *E. grisescens*. Control plants underwent similar treatment in the absence of insect pests. “E” represents groups infested by *E. grisescens*, while “CK” represents groups not infested by *E. grisescens.* CK1, CK2, and CK3 represent three replicates in the control group, while E1, E2, and E3 are three replicates in the treatment group. (**b**) Rarefied fraction curves of soil samples from different treatments at various time intervals. (**c**) Alpha diversity analysis: Chao1 richness index and Shannon_2 diversity index. An asterisk (*) denoted significant difference between treatment and control groups at *p* < 0.05; two asterisk (**) denoted a significant difference at *p* < 0.01. (**d**) Principal Coordinates Analysis (PCoA) of soil bacterial community. (**e**) Relative abundance of major soil bacterial phyla in soil samples.

**Figure 2 insects-16-00412-f002:**
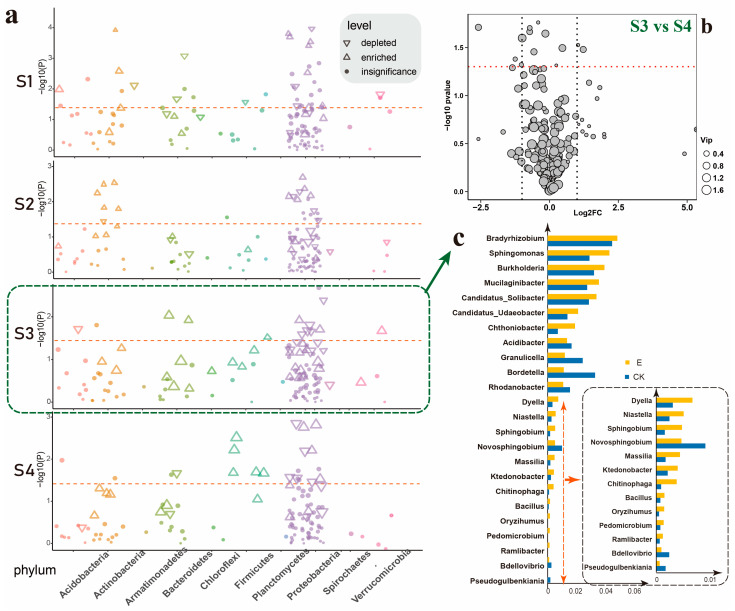
(**a**) Manhattan plot demonstrates the differential genera between the larvae-infested and non-infested groups across various time points (|logFC| > 1). Each point signifies a genus, with the size indicative of the fold change magnitude. Genera within the same phylum are color-coded similarly. (**b**) Volcano plot displayed the distinction in genus-level expression between the larvae-infested group at time points S3 and S4. (**c**) Expression levels of bacterial genera with a relative abundance exceeding 0.01 and exhibiting multiple differences greater than 2 (|logFC| > 1) when comparing the larvae-infested group at time S3 to the non-infested group.

**Figure 3 insects-16-00412-f003:**
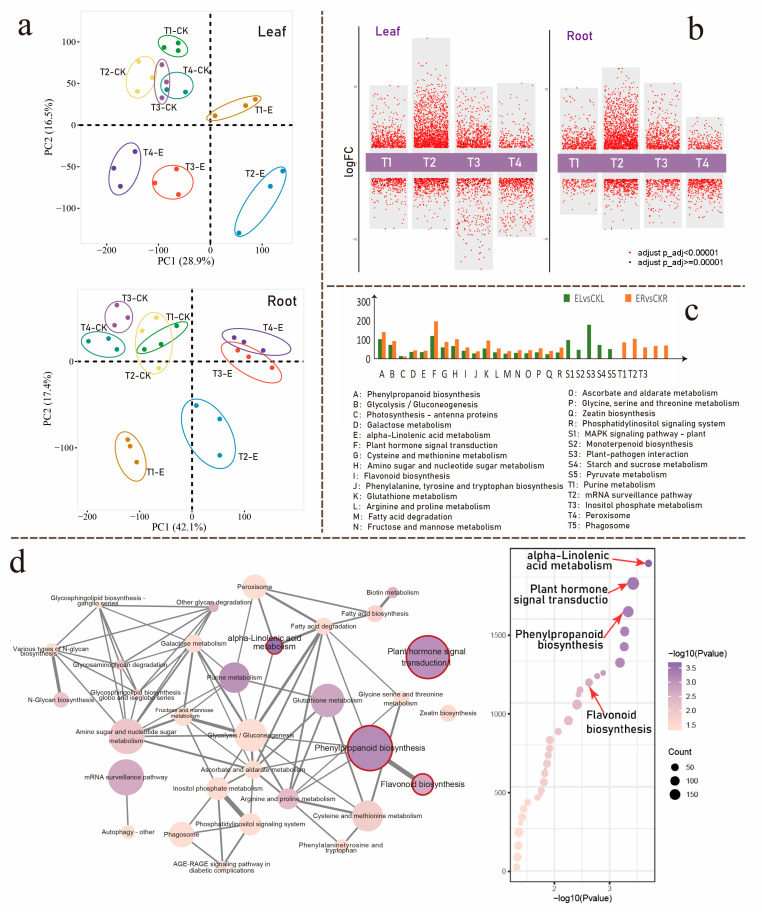
(**a**) Principal Component Analysis (PCA) of sample points for leaves and roots; (**b**) Multi-group difference volcano plots comparing leaf and root systems of the larvae-infested groups vs. non-infested groups at different time points. (**c**) Comparison of KEGG pathways enrichment of DEGs in leaf and root tissues at time point T2; (**d**) Regulation network diagram and bubble plot of KEGG pathways in the root system at time T2. In the network diagram, each node’s size corresponds to the number of differentially expressed genes within that specific pathway, and the node’s color signifies the pathway’s significance level.

**Figure 4 insects-16-00412-f004:**
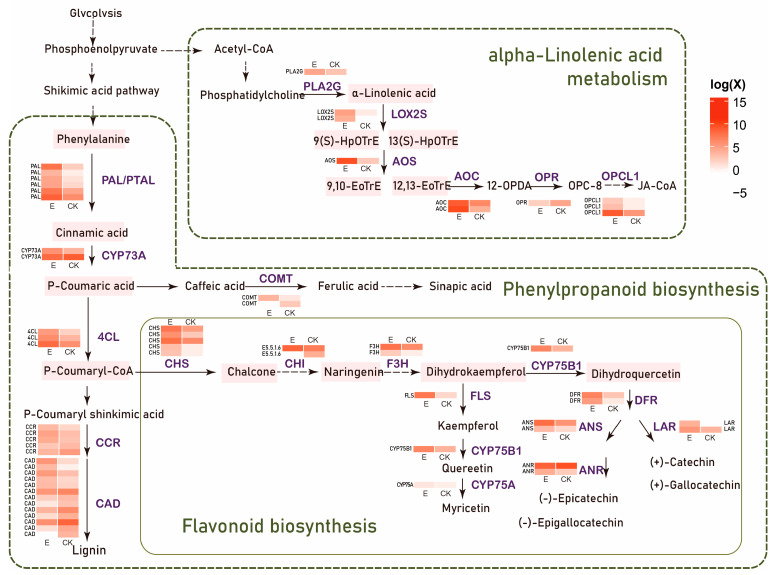
Regulatory networks of three significantly altered secondary metabolic pathways in the root system at time point T2, and the expression levels of corresponding differentially expressed genes.

**Figure 5 insects-16-00412-f005:**
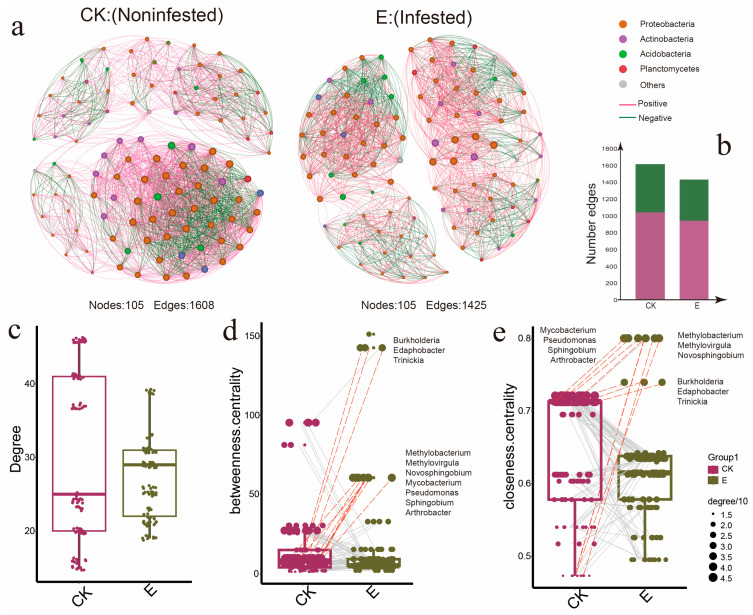
(**a**) Networks of co-occurrence for rhizosphere non-infested or infested with *E. grisescens* Warren. Each genus is represented by a node, with colors corresponding to its phylum. The size of each node is determined by its degree of connection. Positive interactions are indicated by red edges, while negative interactions are represented by green edges. The number of edges (**b**) and degrees (**c**) of rhizosphere networks. The significant difference in degree was assessed by a Kruskal–Wallis test. (**d**) Comparison of node-level topological features in the non-infested and infested networks based on degree and betweenness centrality. (**e**) Comparison of node-level topological features in a non-infested network and an infested network based on the degree and closeness centrality.

**Figure 6 insects-16-00412-f006:**
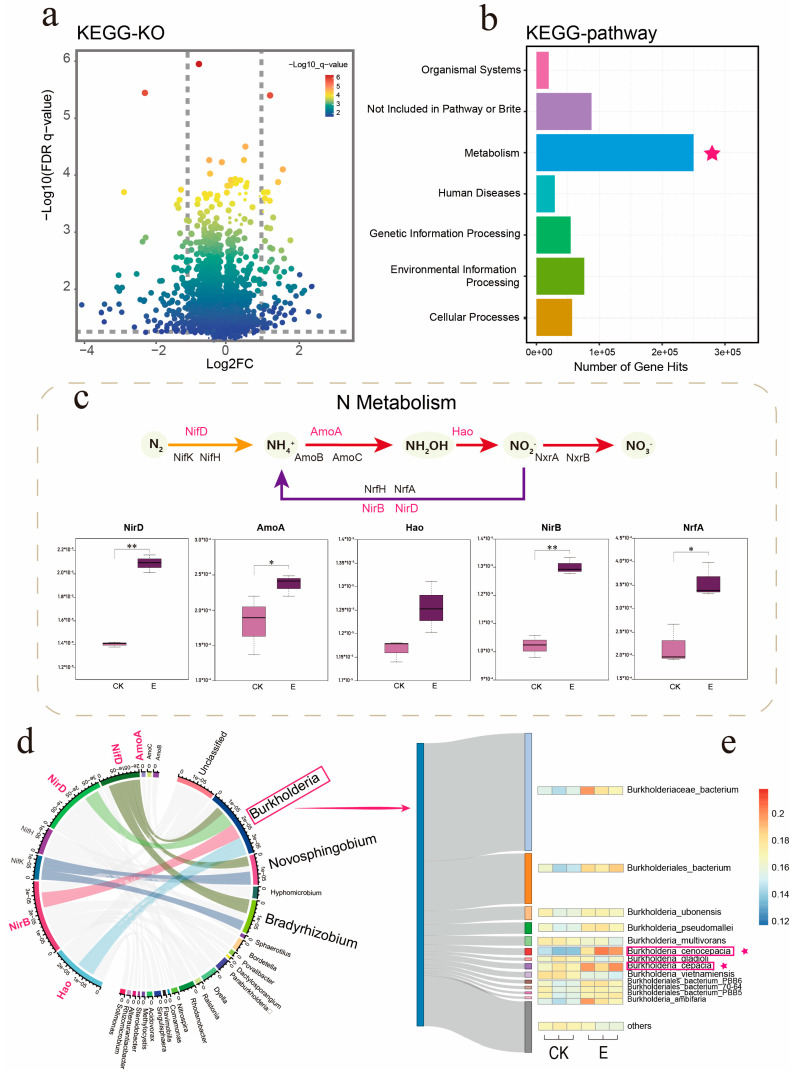
Functional characterization of the rhizosphere microbiome in infested vs. noninfested tea plants using the KEGG database. (**a**) Enriched and depleted KOs in infested plants. (**b**) KEGG pathways with enriched KOs in the rhizosphere microbiome of infested tea plants. (**c**) Differential KOs involved in nitrogen metabolism between the rhizosphere microbiome of infested and noninfested tea plants, highlighting microbial pathways in the nitrogen cycle. The box plot displays expression levels of various genes involved in this process. An asterisk (*) denoted significant difference between treatment and control groups at *p* < 0.05 while two asterisk (**) denoted a significant difference at *p* < 0.01. (**d**) Circos plot illustrating the distribution of nitrogen cycle-related genes within the rhizosphere microbiome, where the width of the line indicates the percentage of distribution. (**e**) Relative abundance of different bacterial species under the genus *Burkholderia* and their expression levels across treatments, represented by the width of the Sankey diagram line, indicating the percentage of distribution.

**Figure 7 insects-16-00412-f007:**
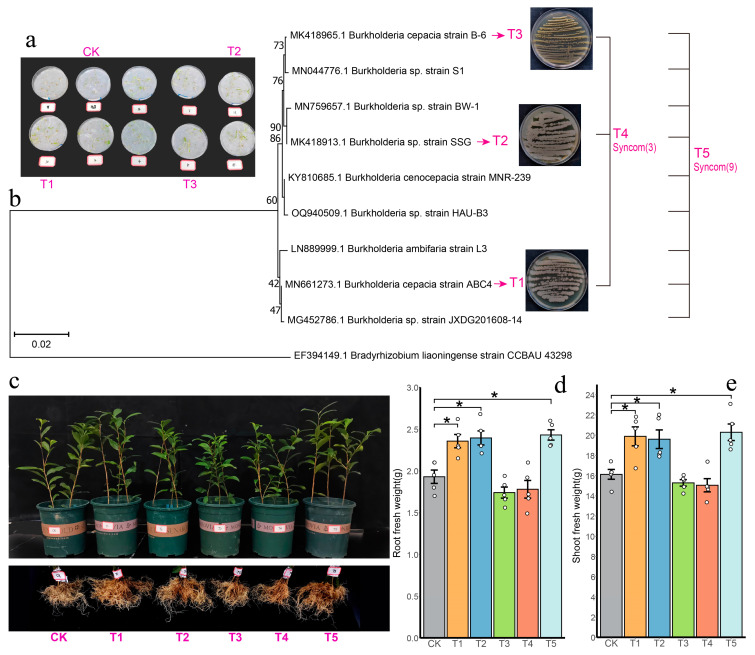
(**a**) Plate growth promotion experiment on tomato seeds treated with various bacterial solutions. (**b**) A phylogenetic analysis of nine *Burkholderia* strains obtained from bacterial plate isolation experiments. The neighbor-joining method was used, and the figures at the nodes indicate bootstrap values (%) derived from 1000 resampled datasets. The NCBI accession numbers for each sequence were provided in parentheses. The bar signified a single nucleotide substitution per 500 nucleotides. (**c**) Growth comparison of aboveground and underground tea seedlings under different treatments. Root fresh weight (**d**) and Shoot fresh weight (**e**) of tea seedlings across different treatments. An asterisk (*) denoted significant difference between treatment and control at *p* < 0.05).

**Figure 8 insects-16-00412-f008:**
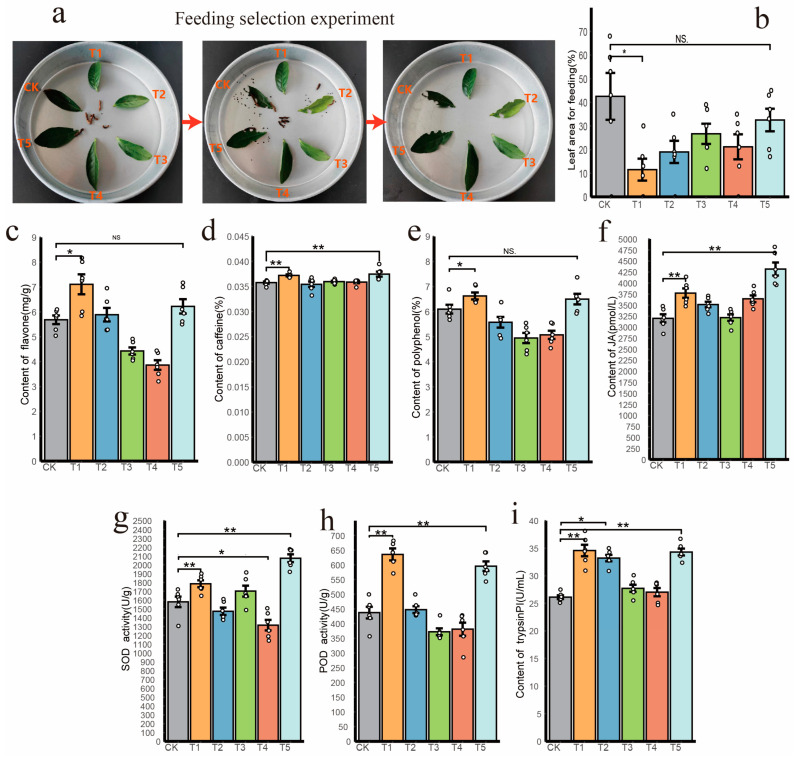
(**a**) Feeding preference experiment of *E. grisescens*. (**b**) Percentage of leaf consumption. (**c**–**i**) Content and enzyme activity of defense compounds under various treatments: Flavonoids content (**c**), Caffeine content (**d**), Polyphenols content (**e**), Jasmonic acid content (JA) (**f**), Trypsin inhibitor content (**i**), SOD activity (**g**), and POD activity (**h**). An asterisk (*) denoted significant difference between treatment and control groups at *p* < 0.05; two asterisk (**) denoted a significant difference at *p* < 0.01 and “NS” indicated that no significant difference at *p* < 0.05.

## Data Availability

The data presented in this study are available on request from the corresponding author due to privacy.

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
