# Peer review of "Impact of *Ectropis grisescens* Warren (Lepidoptera: Geometridae) Infestation on the Tea Plant Rhizosphere Microbiome and Its Potential for Enhanced Biocontrol and Plant Health Management"

_insects, 2025, doi:10.3390/insects16040412_

Round 1

Reviewer 1 Report

Comments and Suggestions for Authors

The study predominantly explores microbiome shifts and plant physiological responses rather than focusing on Ectropis grisescens as an insect pest. While the microbiome aspect is interesting, its direct relevance to entomology and pest control could be better clarified.

If submitting to Insects, the authors may need to better frame their findings within an insect-focused narrative, emphasizing how these microbial shifts influence pest behavior, plant susceptibility, or pest control strategies.

Introduction

The introduction provides a strong foundation by discussing plant defense mechanisms, the role of rhizosphere bacteria, and the importance of studying plant-microbe-insect interactions.

Lines 39–49 "herbivores attacks" → should be "herbivore attacks"

Line 44 "Seeking external support" could be more precise—perhaps specify "attracting beneficial organisms such as predatory insects and microbes."

Line 86 "a significant economic crops" → "a significant economic crop"

Results

The results part presented aim to address the impact of E.  grisescens infestation on the tea rhizosphere microbiome, it presents a comprehensive analysis of microbial community shifts and their potential influence on tea plant responses. Sections 3.1–3.3 provide valuable insights into taxonomic and functional changes in microbial communities, while Sections 3.4–3.5 explore their physiological effects on plants. However, some key aspects require further refinement, particularly in establishing mechanistic links, providing experimental validation, and ensuring cautious interpretation of causality.

3.4. Ectropis grisescens Infestation Affects Tea Rhizosphere Microbiome Function

The details of sequencing, including the total raw reads (439,693,642) and clean reads (348,848,776), are clear. However, more context could be provided on why the quality control step specifically removed plant-derived sequences—was this based on sequence length, quality scores, or contamination?

The annotation of 2,443,384 non-redundant genes from the metagenomic data is significant. However, while the use of KEGG Orthologs (KOs) is common, it’s important to mention how well the annotated KOs align with the specific functions being tested. Were any critical microbial genes unassigned or poorly annotated, and if so, could this affect the interpretation of functional shifts?

Regarding the Microbial Functional Shifts Due to E. grisescens Infestation, I have some questions requires authors give a further explanation:

  1. You report differential enrichment of 115 enriched KOs and 167 depleted KOs with a focus on metabolic pathways. However, this section could benefit from greater interpretation. Are the changes to these pathways directly related to microbial adaptation or are they a response to the plant stress due to infestation?
  2. A detailed description of metabolic pathways is given (e.g., carbohydrate metabolism, amino acid metabolism, nitrogen metabolism). Yet, the biological significance of these enriched pathways is unclear. What are the specific roles of the identified pathways in plant health? Are the changes in pathways driving the plant’s response to pest attack, or do they reflect a shift in the microbiome that facilitates microbial survival under stress conditions?
  3. The section on nitrogen metabolism is well-explained, linking enriched genes like NifD, AmoA, Hao, NirB, and NrfA to nitrogen transformation in the soil. However, there needs to be more direct evidence linking these genes to plant nitrogen uptake. Are the ammonium levels in soil actually higher in infested vs non-infested conditions? If possible, include soil measurements or plant nitrogen content to link microbiome changes to nutrient availability in plants.
  4. The use of a Circos plot to illustrate contributions of microbial genera (Burkholderia, Bradyrhizobium, Novosphingobium) to nitrogen cycling is an excellent visual. However, while the genera involved are identified, what is the mechanistic role of these genera in the nitrogen cycle? For instance, is it nitrogen fixation, denitrification, or ammonification that is mainly affected? This would benefit from a more detailed explanation of why these taxa matter for plant pest resistance.

3.5. Burkholderia Positively Enhances Plant Pest-Resistant Performance

  1. The experiment shows that the T1, T2, and T5 treatments resulted in significant growth improvements in root and shoot weights. This is an important finding. However, could the treatments be compared more clearly to demonstrate the specific role of Burkholderia vs. other potential microbial contributors? Are there any confounding variables (e.g., soil type, other microbial influences)?It would be helpful to clarify the control group (CK) composition. Were these treatments only Burkholderia strains, or was there an additional microbial consortium? This would be important for distinguishing the effects of Burkholderia alone versus the whole microbiome.
  2. The feeding preference experiment provides interesting results, with T1 treatment showing reduced leaf consumption by E. grisescens larvae. This is a significant outcome; however, it would be beneficial to show whether other treatments had any effect on insect preference or whether this result is due to bacterial effects on plant properties (e.g., leaf toughness, chemical composition) rather than microbial presence alone.
  3. The finding that flavonoids, polyphenols, caffeine, and jasmonic acid content increases in the T1 group is well-supported by statistical analysis. However, why these metabolites are involved in pest resistance could be better explained. How do these compounds interact with the pest's feeding behavior or metabolism? Are there any literature references supporting the specific role of these compounds in pest resistance for tea plants?

The Jasmonic acid pathway is often involved in plant defense responses to herbivory, but additional experimental validation showing how Burkholderia strains affect the synthesis or signaling of these metabolites in tea plants would strengthen this conclusion.

Also, the microbiome data and plant response (growth, defense metabolites, pest resistance) are presented separately, but linking them with a direct cause-and-effect pathway will solidify the conclusions. This could be done with experimental manipulations or measuring microbe-plant interactions in more depth (e.g., through plant-specific assays or gene expression profiling).

  1. The analysis of SOD (superoxide dismutase) and POD (peroxidase) in various treatments is a good measure of oxidative stress and defense response. However, it is unclear why T4 showed reduced SOD activity. Does this indicate detrimental effects from a specific Burkholderia strain or an imbalance in the plant's defensive response? Further interpretation of why this decrease is observed in T4 could add more insight into how different treatments modulate plant defenses.

Additionally, trypsin inhibitors are important for direct herbivore deterrence, and the higher levels in T1 and T5 are noteworthy. However, could this correlate to the observed reduced feeding preference of E. grisescens? A clearer link between trypsin inhibitor levels and pest feeding behavior would be useful.

 I would also suggest performing a beta diversity analysis on an overview of microbial community shifts under different infestation conditions. It would be helpful to further clarify the statistical significance of these differences (e.g., effect sizes from PERMANOVA). Additionally, discussing whether specific microbial taxa drive these shifts and linking them to plant or pest responses would strengthen the interpretation. If possible, adding correlation analyses between beta diversity and plant response variables (e.g., root weight, stress markers) could provide more mechanistic insights.

Lastly, the rationale for selecting a 6-hour infestation period should be clarified. Given that microbial shifts and plant responses often take longer to manifest, it would be helpful to provide justification for this timeframe or, if possible, assess changes over an extended period. If 6 hours was chosen based on prior studies or preliminary trials, a brief explanation would strengthen the methodological rigor.

Comments on the Quality of English Language

The manuscript's English is generally understandable but could benefit from improved clarity, conciseness, and grammatical refinement. Some sentences are overly complex or awkwardly structured, affecting readability. I recommend thorough proofreading to enhance fluency, ensure precise terminology, and eliminate minor grammatical errors. 

Author Response

Thank you for reviewing our manuscript and providing some constructive suggestions.  We have carefully considered all your opinions, responded to your questions and suggestions, and made corresponding modifications to the manuscript.

Comments 1:The study predominantly explores microbiome shifts and plant physiological responses rather than focusing on Ectropis grisescens as an insect pest. While the microbiome aspect is interesting, its direct relevance to entomology and pest control could be better clarified.

If submitting to Insects, the authors may need to better frame their findings within an insect-focused narrative, emphasizing how these microbial shifts influence pest behavior, plant susceptibility, or pest control strategies.

Response 1: Thank you for your recognition of our work. We understand your concerns; however, the scope of the journal Insects is not limited solely to pest control but also includes interactions between insects and plants. Our research primarily focuses on the interactions between pests and plant root-associated microorganisms. Additionally, we aim to identify a root microorganism with potential for biological control, thereby pioneering a new approach to biocontrol. In fact, the final part of our experiment also explores the effects of root microorganisms on pest feeding preferences and insect resistance indicators, which are indeed related to pest management.

Comments 2: Lines 39–49 "herbivores attacks" → should be "herbivore attacks"

Response 2: Thank you for your correction. I have changed "herbivores attacks" to "herbivore attacks"

Comments 3: Line 44 "Seeking external support" could be more precise—perhaps specify "attracting beneficial organisms such as predatory insects and microbes."

Response 3: Thank you for your suggestion. I have changed " attracting beneficial organisms such as predatory insects and microbes " to " seek external support " in the text.

Comments 4:Line 86 "a significant economic crops" → "a significant economic crop"

Response 4: Thank you for pointing out this error for us. We have made the correction by changing " a significant economic crops " to " a significant economic crop "

Comments 5: The details of sequencing, including the total raw reads (439,693,642) and clean reads (348,848,776), are clear. However, more context could be provided on why the quality control step specifically removed plant-derived sequences—was this based on sequence length, quality scores, or contamination?

Response 5:Due to the presence of a large number of low-quality bases, adapter contamination, or short lengths in plant derived sequences, these factors can affect the accuracy of subsequent analysis. Therefore, quality checks and filtering were performed on the sequence, resulting in a total of 348848776 clean readings.

Comments 6: You report differential enrichment of 115 enriched KOs and 167 depleted KOs with a focus on metabolic pathways. However, this section could benefit from greater interpretation. Are the changes to these pathways directly related to microbial adaptation or are they a response to the plant stress due to infestation?

Response 6: Thank you for your reminder. I found that I did not clearly explain this issue in the article. The metagenomic sequencing in this article is aimed at soil microorganisms, so the metabolic pathways enriched by differential genes also reflect the adaptation and changes made by soil microorganisms under pest stress in tea plants. I also provided a more detailed description in the original text “It is worth noting that the majority of enriched KOs are associated with metabolic pathways that reflect the adaptation and changes made by soil microorganisms in tea plants under pest stress.”

Comments 7: A detailed description of metabolic pathways is given (e.g., carbohydrate metabolism, amino acid metabolism, nitrogen metabolism). Yet, the biological significance of these enriched pathways is unclear. What are the specific roles of the identified pathways in plant health? Are the changes in pathways driving the plant’s response to pest attack, or do they reflect a shift in the microbiome that facilitates microbial survival under stress conditions?

Response 7: Regarding the significance of identified metabolic pathways, we have added relevant research discussions on carbohydrate metabolism, amino acid metabolism, and nitrogen metabolism in enhancing plant insect resistance or stress resistance in the results section. Line525 “These identified metabolic pathways play important roles in regulating plant growth and defense. Carbohydrate metabolism provides energy and carbon backbone for plant growth[1], amino acid metabolism regulates plant defense signals and antioxidant systems[2,3], and nitrogen metabolism affects the synthesis and distribution of plant defense substances[4,5]. The synergistic effects of various metabolic processes in plants form an energy defense dynamic balance, jointly enhancing their insect resistance and stress resistance.”Regarding whether metabolic changes are driven by plant defense or microbial adaptation issues, I acknowledge that our research has limitations. Currently, our data cannot determine whether it is a plant defense or microbial adaptation issue. Thank you for your valuable feedback, and we can further explore in the future.

Comments 8: The section on nitrogen metabolism is well-explained, linking enriched genes like NifD, AmoA, Hao, NirB, and NrfA to nitrogen transformation in the soil. However, there needs to be more direct evidence linking these genes to plant nitrogen uptake. Are the ammonium levels in soil actually higher in infested vs non-infested.

Response 8: Thank you very much for your reminder and suggestion. By measuring the true ammonium content in soil, direct evidence can be provided to link nitrogen cycle related genes with plant nitrogen absorptionAlthough we did not specifically measure the ammonium level in soil, we found in subsequent experiments (not yet published) that the nitrogen metabolism pathway in plant roots was significantly upregulated, indicating that the upregulation of these genes improves plant nitrogen absorption, and there is a direct connection between the two. According to your proposal, we have provided additional explanations in Line 538 of this article “And in our subsequent experiments, we measured the transcriptome of plant roots and found that the nitrogen metabolism pathway in roots was significantly upregulated(not yet published), indicating that upregulation of these genes can improve plant nitrogen absorption and transformation.”

Comments 9: conditions? If possible, include soil measurements or plant nitrogen content to link microbiome changes to nutrient availability in plants.

Response 9: Thank you for your suggestion. In response to this issue, we believe that this article focuses on exploring the impact of microbial inoculants on the improvement of plant insect resistance performance. Moreover, the nitrogen nutrition in plants can only be better utilized in the synthesis of insect resistant substances to better demonstrate the effectiveness of microbial inoculants in improving plant insect resistance performance. Therefore, by measuring some nitrogen-containing compounds such as caffeine, polyphenols, and trypsin inhibitors in plant leaves, it was found that the level of insect resistant substances increased, which is sufficient to illustrate this issue. So, we believe that further determination of soil and plant nitrogen content is unnecessary.

Comments 10: The use of a Circos plot to illustrate contributions of microbial genera (Burkholderia, Bradyrhizobium, Novosphingobium) to nitrogen cycling is an excellent visual. However, while the genera involved are identified, what is the mechanistic role of these genera in the nitrogen cycle? For instance, is it nitrogen fixation, denitrification, or ammonification that is mainly affected? This would benefit from a more detailed explanation of why these taxa matter for plant pest resistance.

Response 10: Thank you for pointing out the shortcomings in this section. In response to the lack of detailed description in this section, we have provided additional explanations in Line 730 below, this can better reveal the contributions of these bacterial genera to soil and plants “Burkholderia, Bradyrhizobium and Novosphingobium are all soil nitrogen cycling related microorganisms that contribute many related genes in the process of soil nitrogen cycling. Burkholderia[6,7] and Bradyrhizobium[8,9] convert nitrogen in the air into ammonia through nitrogen fixation, providing important nitrogen sources for plants and promoting their growth and development. In addition, Novoschingbium[10] participate in ammonification, decomposing organic nitrogen compounds into ammonia, providing additional nitrogen sources for plants and promoting nitrogen cycling.”

Comments 11: The experiment shows that the T1, T2, and T5 treatments resulted in significant growth improvements in root and shoot weights. This is an important finding. However, could the treatments be compared more clearly to demonstrate the specific role of Burkholderia vs. other potential microbial contributors? Are there any confounding variables (e.g., soil type, other microbial influences)?It would be helpful to clarify the control group (CK) composition. Were these treatments only Burkholderia strains, or was there an additional microbial consortium? This would be important for distinguishing the effects of Burkholderia alone versus the whole microbiome.

Response 11: During the bacterial inoculation process, the same soil was used for 5 treatment groups and 1 control group, and the soil used was sterilized. Therefore, there were no confounding variables such as soil type or other microorganisms that interfered with this experiment.

Comments 12: The feeding preference experiment provides interesting results, with T1 treatment showing reduced leaf consumption by E. grisescens larvae. This is a significant outcome; however, it would be beneficial to show whether other treatments had any effect on insect preference or whether this result is due to bacterial effects on plant properties (e.g., leaf toughness, chemical composition) rather than microbial presence alone.

Response 12:The differences in feeding preferences of the E. grisescens are mainly due to the varying effects of different bacterial treatments on plant characteristics. The tea seedling leaves treated with T1 were least consumed by the E. grisescens, and the expression levels of insect resistant compounds in tea seedling leaves were subsequently measured. The results showed that the levels of insect resistant compounds were higher in the T1 treatment. Therefore, the experimental results of the feeding preference of the E. grisescens are consistent with the expression levels of insect resistant compounds in tea seedling leaves, and also reveal that the feeding preference of the E. grisescens is mainly influenced by different bacterial treatments on plant characteristics. In response to your suggestion, I have also provided further explanation in the article. Line785 “We conducted feeding experiments with E. grisescens, and the results showed that the leaves of tea seedlings treated with T1 were consumed the least by the E. grisescens. Subsequently, the expression levels of insect resistant compounds in tea seedling leaves were measured, and the results showed that the levels of insect resistant compounds were higher in the T1 treatment. Therefore, the experimental results of the feeding preference of the E. grisescens are consistent with the expression levels of insect resistant compounds in tea seedling leaves, and also reveal that the feeding preference of the E. grisescens is mainly influenced by different bacterial treatments on plant chemical characteristics. Further indicating that T1 treatment enhances the level of insect resistant compounds in tea seedling leaves after inoculation with bacteria, thereby affecting the feeding behavior of the E. grisescens. ” However, regarding the impact of leaf structure on pest feeding that you mentioned, we did not pay attention to this aspect. Thank you for your valuable feedback, and we will pay attention to this issue in future experiments.

Comments 13: The finding that flavonoids, polyphenols, caffeine, and jasmonic acid content increases in the T1 group is well-supported by statistical analysis. However, why these metabolites are involved in pest resistance could be better explained. How do these compounds interact with the pest's feeding behavior or metabolism? Are there any literature references supporting the specific role of these compounds in pest resistance for tea plants?

Response 13: Thank you for pointing out the shortcomings of the article. We did not provide a detailed introduction to explain the effects of these substances. We are glad that you helped us point them out. We have supplemented the article to further explain the effects of these enhanced compounds on the feeding behavior of pests. Line770 “In addition, the increased polyphenols, flavonoids and caffeine, as constitutive defense compounds in plants, affect the feeding, growth, and survival of herbivores[11]. These substances have a strong bitter taste, which can affect the oral muscle activity of herbivores and reduce the feeding efficiency of pests[12]. Polyphenols can bind to digestive enzymes in the digestive tract of pests, reducing their digestive capacity and affecting their growth and development[13]. Caffeine has neurotoxicity to insects, which can affect their behavior and survival ability[12].”

Comments 14: The Jasmonic acid pathway is often involved in plant defense responses to herbivory, but additional experimental validation showing how Burkholderia strains affect the synthesis or signaling of these metabolites in tea plants would strengthen this conclusion.

Response 14: Thank you for your valuable suggestion. While we agree that elucidating the mechanistic link between Burkholderia and jasmonic acid (JA) signaling would further strengthen the conclusions, our current study focuses on identifying pest resistance-associated soil microbiomes with potential for biocontrol applications. Although we did not perform direct experimental validation of microbial-JA interactions, we measured JA concentrations in tea leaves and observed a positive correlation with Burkholderia type (data included in the manuscript). This supports their potential role in plant defense, consistent with prior studies linking Burkholderia to JA pathway modulation. Further mechanistic validation falls beyond the screening-focused scope of this work but will be prioritized in follow-up studies.

Comments 15:Also, the microbiome data and plant response (growth, defense metabolites, pest resistance) are presented separately, but linking them with a direct cause-and-effect pathway will solidify the conclusions. This could be done with experimental manipulations or measuring microbe-plant interactions in more depth (e.g., through plant-specific assays or gene expression profiling).

Response 15: Thank you for your insightful comments. We agree that establishing causal links between microbial shifts and plant responses would deepen the mechanistic understanding. However, our study is designed as a two-fold exploration: (1) observing tea plant-microbe interactions under pest stress, and (2) preliminarily identifying soil microbes with potential for biocontrol applications without overextending into mechanistic plant-microbe interplay.

Comments 16:The analysis of SOD (superoxide dismutase) and POD (peroxidase) in various treatments is a good measure of oxidative stress and defense response. However, it is unclear why T4 showed reduced SOD activity. Does this indicate detrimental effects from a specific Burkholderia strain or an imbalance in the plant's defensive response? Further interpretation of why this decrease is observed in T4 could add more insight into how different treatments modulate plant defenses.

Response 16: Thank you for your suggestion. The reason for the decrease in SOD activity under T4 treatment may be due to the interaction between mixed Burkholderia strains or the imbalance of plant defense responses。Our article has a lot of main content, and the exploration of the reasons for the decrease in SOD activity under T4 treatment is not within our main content. To explore these mechanisms, more experiments and analysis are needed, But thank you for your suggestion. We will further explore this part in our future research.

Comments 17:Additionally, trypsin inhibitors are important for direct herbivore deterrence, and the higher levels in T1 and T5 are noteworthy. However, could this correlate to the observed reduced feeding preference of E. grisescens? A clearer link between trypsin inhibitor levels and pest feeding behavior would be useful.

Response 17:Thank you for your suggestion. We have supplemented the evidence of trypsin inhibitor insect resistance in the discussion section and added relevant content to Line766. “Many studies have revealed the adverse effects of trypsin inhibitors on pests[14,15]. Trypsin inhibitors bind to trypsin in the digestive tract of insects to form enzyme inhibitor complexes (EI), which block or weaken the hydrolysis of food proteins by trypsin, leading to obstruction of insect nutrient absorption and interference with their growth and development.”

Comments 18:Lastly, the rationale for selecting a 6-hour infestation period should be clarified. Given that microbial shifts and plant responses often take longer to manifest, it would be helpful to provide justification for this timeframe or, if possible, assess changes over an extended period. If 6 hours was chosen based on prior studies or preliminary trials, a brief explanation would strengthen the methodological rigor.

Response 18: Thank you for your suggestion. The pest infestation time of 6 hours is determined based on the extent to which insects feed on leaves.According to previous literature records, the optimal treatment time is around one-third of the leaf area for insects to feed on. The time from feeding to removing insects is 6 hours. “According to previous literature records[16], the optimal treatment time is around one-third of the leaf area for insects to feed on. The larvae were then allowed to feed for 6 hours, take 1/3 of the leaf area, after which they were removed.”

Comments 19:Comments on the Quality of English Language

The manuscript's English is generally understandable but could benefit from improved clarity, conciseness, and grammatical refinement. Some sentences are overly complex or awkwardly structured, affecting readability. I recommend thorough proofreading to enhance fluency, ensure precise terminology, and eliminate minor grammatical errors. 

Response 19: Regarding the language suggestion you mentioned, we plan to have a native English speaker polish it after the final draft is finalized.

References

  1. Aguirre, M.; Kiegle, E.; Leo, G.; Ezquer, I. Carbohydrate reserves and seed development: An overview. Plant reproduction 2018, 31, 263-290.
  2. Guo, Z.; Gong, J.; Luo, S.; Zuo, Y.; Shen, Y. Role of gamma-aminobutyric acid in plant defense response. Metabolites 2023, 13, 741.
  3. Sharma, S.S.; Dietz, K.-J. The significance of amino acids and amino acid-derived molecules in plant responses and adaptation to heavy metal stress. Journal of experimental botany 2006, 57, 711-726.
  4. Zayed, O.; Hewedy, O.A.; Abdelmoteleb, A.; Ali, M.; Youssef, M.S.; Roumia, A.F.; Seymour, D.; Yuan, Z.-C. Nitrogen journey in plants: From uptake to metabolism, stress response, and microbe interaction. Biomolecules 2023, 13, 1443.
  5. Zhang, J.; Lv, J.; Xie, J.; Gan, Y.; Coulter, J.A.; Yu, J.; Li, J.; Wang, J.; Zhang, X. Nitrogen source affects the composition of metabolites in pepper (Capsicum annuum L.) and regulates the synthesis of capsaicinoids through the GOGAT–GS pathway. Foods 2020, 9, 150.
  6. Pang, Z.; Mao, X.; Zhou, S.; Yu, S.; Liu, G.; Lu, C.; Wan, J.; Hu, L.; Xu, P. Microbiota-mediated nitrogen fixation and microhabitat homeostasis in aerial root-mucilage. Microbiome 2023, 11, 85.
  7. Estrada-De Los Santos, P.; Bustillos-Cristales, R.o.; Caballero-Mellado, J.s. Burkholderia, a genus rich in plant-associated nitrogen fixers with wide environmental and geographic distribution. Applied and environmental microbiology 2001, 67, 2790-2798.
  8. Zimmer, S.; Messmer, M.; Haase, T.; Piepho, H.-P.; Mindermann, A.; Schulz, H.; Habekuß, A.; Ordon, F.; Wilbois, K.-P.; Heß, J. Effects of soybean variety and Bradyrhizobium strains on yield, protein content and biological nitrogen fixation under cool growing conditions in Germany. European Journal of Agronomy 2016, 72, 38-46.
  9. Hungria, M.; Menna, P.; Delamuta, J.R.M. Bradyrhizobium, the ancestor of all rhizobia: Phylogeny of housekeeping and nitrogen‐fixation genes. Biological nitrogen fixation 2015, 191-202.
  10. Huang, X.; Zhou, W.; Zhang, Y.; Yang, Q.; Yang, B.; Liang, T.; Ling, J.; Dong, J. Keystone PGPR ecological effect: An inoculation case study of diazotrophic Novosphingobium sp. N034 on mangrove plant Kandelia obovate. Applied Soil Ecology 2024, 202, 105567.
  11. Ramaroson, M.-L.; Koutouan, C.; Helesbeux, J.-J.; Le Clerc, V.; Hamama, L.; Geoffriau, E.; Briard, M. Role of phenylpropanoids and flavonoids in plant resistance to pests and diseases. Molecules 2022, 27, 8371.
  12. Ceja-Navarro, J.A.; Vega, F.E.; Karaoz, U.; Hao, Z.; Jenkins, S.; Lim, H.C.; Kosina, P.; Infante, F.; Northen, T.R.; Brodie, E.L. Gut microbiota mediate caffeine detoxification in the primary insect pest of coffee. Nature communications 2015, 6, 7618.
  13. Singh, S.; Kaur, I.; Kariyat, R. The multifunctional roles of polyphenols in plant-herbivore interactions. International Journal of Molecular Sciences 2021, 22, 1442.
  14. Graham, J.; McNicol, R.; Greig, K. Towards genetic based insect resistance in strawberry using the cowpea trypsin inhibitor gene. Annals of Applied Biology 1995, 127, 163-173.
  15. Boulter, D.; Gatehouse, A.; Hilder, V. Use of cowpea trypsin inhibitor (CpTI) to protect plants against insect predation. Biotechnology advances 1989, 7, 489-497.
  16. Wang, Y.-N.; Tang, L.; Hou, Y.; Wang, P.; Yang, H.; Wei, C.-L. Differential transcriptome analysis of leaves of tea plant (Camellia sinensis) provides comprehensive insights into the defense responses to Ectropis oblique attack using RNA-Seq. Functional & integrative genomics 2016, 16, 383-398.

Reviewer 2 Report

Comments and Suggestions for Authors

In the manuscript insects-3517371, Liu et al. integrated transcriptomic and metagenomic analyses with bacterial feedback assays to investigate the temporal dynamics of the root-associated microbiome and its response to Ectropis grisescens infestation. Their study focuses on plant defense mechanisms and microbial community changes, providing insightful findings with valuable implications for sustainable pest management in tea cultivation. The topic is engaging, and the experiments and analyses appear well-executed. I recommend accepting the manuscript after addressing the following detailed comments:

Line 16 Remove the comma after "roots."

Line 16-19 The sentence is too long; consider splitting it for better readability.

Line 21 Insert a missing space before "By"

Line 21 Change "16SrRNA" to "16S rRNA"

Line 32 Provide the full names of "SOD" and "POD" when first mentioned.

Line 44 The introduction section is detailed, but missing references after 'seeking external support to combat the stress', please supplement.

Line 86 Change "crops" to "crop"

Line 102 The term “fluorescent Pseudomonad strains” here does not belong to the genus of bacteria and cannot be italicized

Line 144 Did pest infestation control environmental variables in different treatments? If there is anything, it should be added

Line 153 Change "uninfested" to "non-infected". Please check the entire text for consistency.

Line 270 “petri dish” should be changed to “Petri dish”

Line 313 Insert a space between “software” and “(version 1.7.0)”.

Line 340, 341 The spaces in 'larvae - inflated' should be removed

Lines 354 Remove the extra spaces.

Lines 362 Insert a space after "E." in "E.grisescens"
Line 378 Delete the hyphen ("-").
Line 415 Italicize "P"
Line 496 Insert a space after "networks." and before "The"
Line 551 Correct "procoess?" to "process" (if this is a typo).
Line 564 Change "treatment" to "treatments."

Line848 References 27 and 29 have duplicates.

Many scientific names in the paper titles of references are not italicized. Please verify one by one.

Fig 3b: The figure should clearly indicate whether it shows "larvae-infested groups vs non-infested group" or "non-infested groups vs larvae-infested group."

Author Response

We would like to express our sincere gratitude to the reviewers for their meticulous review and constructive feedback, which has given us theopportunity to improve our manuscript in various aspects. We have carefullyaddressed each comment and have made modifications accordingly.Thank you again for summarizing the manuscript and providing encouraging comments.

Comments 1:Line 16 Remove the ',' after 'roots'
Response 1: Thank you for your reminder. I have removed the unnecessary ','

Comments 2:Line 16-19 is too long, I suggest splitting it for easier understanding
Response 2:“Line 16-19” has been changed to“The study took samples of leaves, roots and rhizosphere soil at different times after the plants were attacked by E. grisescens. These samples were analyzed using transcriptomic and high-throughput sequencing of 16S rRNA. techniques. The goal was to understand how the plant's defense mechanisms and the microbial community around the roots changed after the attack.”

Comments 3:Line 21 Missing space before 'By'
Response 3: Thank you for the reminder. A space has been added before 'By'.

Comments 4:Line 21 Change '16SrRNA' to '16S rRNA'
Response 4: I have changed '16SrRNA' to '16S rRNA.

Comments 5:Line 33 The full names of "SOD" and "POD" should be used when they first appear in the text
Response 5:Thank you very much for helping me find these errors, I have supplemented the entire process of "SOD" and "POD", changed "POD and SOD" in the original text to 'Peroxidase(POD) and Superoxide Dismutase(SOD)'.

Comments 6:Line 45 The introduction section is detailed, but missing references after 'seeking external support to combat the stress', please supplement.
Response 6: For the introduction section, we have added some latest references to support the article,including [2-6]

[2] Heil, M. Herbivore‐induced plant volatiles: targets, perception and unanswered questions. 2014.

[3] Dicke, M.; Baldwin, I.T. The evolutionary context for herbivore-induced plant volatiles: beyond the ‘cry for help’. Trends in plant science 2010, 15, 167-175.

[4] Turlings, T.C.; Erb, M. Tritrophic interactions mediated by herbivore-induced plant volatiles: mechanisms, ecological relevance, and application potential. Annual review of entomology 2018, 63, 433-452.

[5] Li, L.; Li, T.; Jiang, Y.; Yang, Y.; Zhang, L.; Jiang, Z.; Wei, C.; Wan, X.; Yang, H. Alteration of local and systemic amino acids metabolism for the inducible defense in tea plant (Camellia sinensis) in response to leaf herbivory by Ectropis oblique. Archives of Biochemistry and Biophysics 2020, 683, 108301.

[6] Wu, J.; Baldwin, I.T. New insights into plant responses to the attack from insect herbivores. Annual review of genetics 2010, 44, 1-24.

Comments 7:Line 87: Change "crops" to "crop."
Response 7: Thank you for pointing out the error,I have changed "crops" to "crop."

Comments 8:Line 105 The term 'fluorescent Pseudomonad strains' here does not belong to the genus of bacteria and cannot be italicized
Response 8:Thank you for your reminder. I have realized this mistake. 'Pseudomonas' does not belong to the bacterial genus, and I have now changed the word 'Pseudomonas' to non italic.

Comments 9:Line 149 Did pest infestation control environmental variables in different treatments? If there is anything, it should be added
Response 9: During the entire experimental process, we controlled for environmental variables, We have added in the article Line154 “Control the environment variables of each processing group to be the same”

Comments 10:Line 155: Change "uninfested" to "non-infected." Please check the entire text for consistency.
Response 10: Thank you for your reminder. I have corrected all inappropriate use of "uninfested" in the entire text to "non-infected".

Comments 11:Line 277 'petri dish' should be changed to 'Petri dish'
Response 11: Thank you for your correction. I have changed 'petri dish' to 'Petri dish'.

Comments 12:Line 315: Insert a space between "software" and "(version 1.7.0)."
Response 12: Thank you for your reminder. I think what you pointed out should be Line 320, and I have inserted a space between 'software' and '(version 1.7.0)'.

Comments 13:Line 348 and Line 349 The spaces in 'larvae - inflated' should be removed

Response 13:Thank you for reminding me. I just noticed this issue and have now removed the space in 'larvae - inflated' .

Comments 14:Lines 357, 593: Remove the extra spaces.
Response 14: I checked Lines 357 and 593 and did not find any extra spaces.

Comments 15:Insert a space after "E." in "E.grisescens"
Response 15:I have inserted a space after "E." in "E.grisescens", changed "E.grisescens" to "E. grisescens" and checked all the "E.grisesens" in the entire text.

Comments 16:Line 385: Delete the hyphen ("-").
Response 16: Thank you for your reminder. I have removed the extra hyphens.

Comments 17:Lines 384-387, 716: Change the commas to non-italicized font.
Response 17: I have changed the commas to non-italicized font. Change the original text to "Bradyrhizobium, Burkholderia-Caballeronia-Paraburkholderia, Sphingomonas, Mucilaginibacter, Candidatus Solibacter, and Candidatus Udaeobacter. In addition, Chthoniobacter, Dyella, Niatella, Massilia, Ktedonobacter, Sphingobium, Bdellovibrio, Pedomicrobium, Oryzihumus, Ramlibactor and Bacillus " .

Comments 18:Line 419: Italicize "P."
Response 18: Thank you for your reminder. I did not find the italicized "P" at line 419. Perhaps you would like to point it out as the "P" at line 428, so I italicized the "P" at line 428.

Comments 19:Line 504: Insert a space after "networks." and before "The."
Response 19:I have inserted a space after "networks." and before "The."

Comments 20:Line 529: Delete the extra "the."
Response 20:I have deleted the extra "the."

Comments 21:Line 560: Correct "procoess?" to "process" (if this is a typo).
Response 21: This is indeed a writing error. Thank you for pointing it out. I have corrected it to "process".

Comments 22:Line 572: Change "treatment" to "treatments."
Response 22: I haved changed "treatment" to "treatments."

Comments 23:Line885 References 27 and 29 have duplicates.
Response 23: Thank you for your reminder. I just noticed this issue and have now reinserted the literature.

Comments 24:Many scientific names in the paper titles of references are not italicized. Please verify one by one.
Response 24: Thank you for pointing out this error. I have rechecked the scientific names in the paper titles of the references and italicized the content that needs to be italicized.

Comments 25:Fig 3b: The figure should clearly indicate whether it shows "larvae-infested groups vs non-infested group" or "non-infested groups vs larvae-infested group."
Response 25: Regarding "Fig 3b", I have made clearer corrections to the title and changed the original content to "(b) Multi-group difference volcano plots comparing leaf and root systems of the larvae-infested groups vs non-infested groups at different time points."

Round 2

Reviewer 1 Report

Comments and Suggestions for Authors

I appreciate the author's efforts in revising and improving the manuscript based on our suggestions. The revisions have addressed the key concerns, and I believe the manuscript is now in a strong position for publication. I would be happy to see it accepted.